# Surface triggered stabilization of metastable charge-ordered phase in SrTiO$_3$

Kitae Eom [1,2,12], Bongwook Chung [1,12], Sehoon Oh[1], Hua Zhou [3], Jinsol Seo[4], Sang Ho Oh [4], Jinhyuk Jang [5], Si-Young Choi [5], Minsu Choi[1], Ilwan Seo [6], Yun Sang Lee[6], Youngmin Kim[7], Hyungwoo Lee[7,8], Jung-Woo Lee [9], Kyoungjun Lee[10], Mark Rzchowski[11], Chang-Beom Eom[10] ✉ & Jaichan Lee [1] ✉

Charge ordering (CO), characterized by a periodic modulation of electron density and lattice distortion, has been a fundamental topic in condensed matter physics, serving as a potential platform for inducing novel functional properties. The charge-ordered phase is known to occur in a doped system with high $d$-electron occupancy, rather than low occupancy. Here, we report the realization of the charge-ordered phase in electron-doped (100) SrTiO$_3$ epitaxial thin films that have the lowest $d$-electron occupancy i.e., $d^1$-$d^0$. Theoretical calculation predicts the presence of a metastable CO state in the bulk state of electron-doped SrTiO$_3$. Atomic scale analysis reveals that (100) surface distortion favors electron-lattice coupling for the charge-ordered state, and triggering the stabilization of the CO phase from a correlated metal state. This stabilization extends up to six unit cells from the top surface to the interior. Our approach offers an insight into the means of stabilizing a new phase of matter, extending CO phase to the lowest electron occupancy and encompassing a wide range of 3$d$ transition metal oxides.

Strongly correlated materials are a fascinating platform to explore emergent phenomena because the macroscopic properties are often governed by a collective behavior of a charge, lattice, spin, and orbital structures rather than by its individual components. One prominent example, charge ordering (CO) refers to the ordering of the metal ions in different oxidation states in specific lattice sites of mixed valent materials, resulting in the system into insulating or semiconducting properties. The CO phase often hides in the phase diagrams and is usually competing with superconductivity[1–3] or mediating the colossal magnetoresistance[1,4–6], thus has been the fundamental subject in condensed matter physics. Furthermore, the nature of symmetry breaking in the CO phase enables to induce of ferroelectricity, even multiferroicity[7–9], making it more attractive for device applications. While the precise microscopic mechanism of CO formation remains under intense scrutiny, the presence of intrinsic electronic instabilities concomitant with structural distortion is widely accepted for the critical boundary condition to stabilize CO phase[10,11]. Therefore, a predictive guideline for the stabilization of CO is of great interest both from fundamental and technological points of view.

[1]School of Advanced Materials Science and Engineering, Sungkyunkwan University (SKKU), Suwon 16419, Republic of Korea. [2]Department of Electronic Engineering, Gachon University, Seongnam 13120, Republic of Korea. [3]X-ray Science Division, Advanced Photon Source, Argonne National Laboratory, Lemont, IL 60439, USA. [4]Department of Energy Engineering, KENTECH Institute for Energy Materials and Devices, Korea Institute of Energy Technology (KENTECH), Naju 58330, Republic of Korea. [5]Department of Materials Science and Engineering, Pohang University of Science and Technology (POSTECH), Pohang, Gyeongbuk 37673, Republic of Korea. [6]Department of Physics and Integrative Institute of Basic Sciences, Soongsil University, Seoul 06978, Republic of Korea. [7]Department of Energy Systems Research, Ajou University, Suwon 16499, Republic of Korea. [8]Department of Physics, Ajou University, Suwon 16499, Republic of Korea. [9]Department of Materials Science and Engineering, Hongik University, Sejong 30016, Republic of Korea. [10]Department of Materials Science and Engineering, University of Wisconsin-Madison, Madison, WI 53706, USA. [11]Department of Physics, University of Wisconsin, Madison, WI 53706, USA. [12]These authors contributed equally: Kitae Eom, Bongwook Chung. ✉e-mail: ceom@wisc.edu; jclee@skku.edu

In this work, we report the discovery of a charge-ordered phase in epitaxial thin films of strongly correlated oxide with the lowest $d$-electron occupancy. As a model system for this study, we used electron-doped $SrTiO_3$ (STO), that is 25% La-doped $SrTiO_3$ (LSTO) which is far below the onset of the Mott transition, following a Fermi-liquid behavior. Strontium titanate has been placed at the center of oxide electronics yielding an immense variety of physical properties including superconductivity[12], ferroelectricity[13], flexoelectricity[14], and ferromagnetism[15,16]. Among them, the nature of superconductivity in STO has been a longstanding subject of debate in the literature[17–19] although it was the first oxide superconductor to be discovered[12]. In particular, recent research has shown that the CO phases encroach on the superconducting phase in other families of oxide superconductors such as cuprates and nickelate[20,21]. Therefore, a natural question is whether these ordered states are also present in STO. Here, we propose a general framework for creating a CO phase in perovskite oxides.

## Results and discussion

To test the feasibility of the CO phase in the electron-doped STO, we first perform first-principles density functional theory (DFT) calculations and find the presence of an insulating charge-ordered state. This metastable state shows the periodic expansion and contraction of the oxygen octahedral pattern, distinct from the metallic nature of bulk LSTO state (Fig. 1a). It also shows the periodic charge modulation of $Ti^{3+}O_6$ and $Ti^{4+}O_6$ with the expanded and contracted $TiO_6$ octahedra (Fig. 1b) and exhibits the antiferromagnetic Mott insulating behavior with the localized $d$ band (Supplementary Fig. 1). To explore a possible way to stabilize metastable CO phase, we further analyze the lattice modulation by decomposing it into the 4 representative octahedral distortion modes: the inter-layer breathing mode ($\Phi_1$), intra-layer breathing mode ($\Phi_2$), Jahn-Teller type distortion mode ($\Phi_3$), and the antiferrodistortive rotation ($\Phi_{z^-}$). (see Fig. 1c–f, and Supplementary Note 1). Analyzed oxygen displacements and oxygen octahedral rotation angle of each distortion mode at the CO phase are $d_1 = 4.24$ pm, $d_2 = 3.41$ pm, $d_3 = 2.59$ pm, and $\theta_{z^-} = 9.08°$, respectively. Since the amplitude of the inter-layer breathing mode ($\Phi_1$) is relatively larger than that of other distortion modes, we hypothesize that the activation of the inter-layer breathing mode ($\Phi_1$) may be a critical role in stabilizing the CO phase.

In the thin film geometry, the surface distortion is usually initiated by the termination of a crystalline material along the surface normal direction, i.e., dangling bonds at the topmost surface. For example, considering a single-terminated perovskite oxide surface, the relaxations of shortening the bond lengths generally result in the displacement of surface anions away from the surface and that of surface cations towards the bulk[22–24]. Especially, the apical oxygen of the octahedron just below the surface is pulled up towards the surface (Fig. 1g), which is similar to the oxygen motion of inter-layer breathing mode ($\Phi_1$) (Fig. 1c). This is due to the decreased coordination number of Ti cations i.e., from $TiO_6$ octahedra to $TiO_5$ square pyramids. This indicates that the distortion occurring at the single terminated (001) oriented surface can be a driving force to promote the breathing mode along the surface normal direction, triggering stabilization of the CO phase. In particular, when the film thickness reaches the ultrathin regime, one expects strong modification of the properties of the entire film such as insulating properties which are macroscopic properties of the CO phase. Therefore, we conclude that the metastable CO phase can be stabilized in (001) oriented thin film of electron-doped STO.

To activate the required inter-layer breathing mode ($\Phi_1$) along the surface normal direction, we experimentally synthesized high-quality $La_{0.25}Sr_{0.75}TiO_3$ (LSTO) epitaxial thin films using pulsed laser deposition on $TiO_2$ terminated (001) oriented STO single crystal substrate (see Supplementary Note 2). The film thicknesses range from 4 to 30 unit cells (u.c.), which is controlled by in-situ reflection high energy electron diffraction (Supplementary Fig. 4). A θ−2θ X-ray diffraction analysis of the films shows little change in the out-of-plane lattice constant as a function of LSTO film thicknesses (Fig. 2a). The in-plane lattice parameter of the LSTO films is coincident with the STO substrate, indicating that the LSTO films are coherently strained (Fig. 2b). It should be noted that lattice mismatch between STO and LSTO is negligible[25], thus a strain is not likely to be the origin of our observation. The contrast inverse annular bright-field (ABF) STEM image shows that LSTO/STO interfaces are atomically sharp and do not have any misfit dislocations (Fig. 2c).

To explore the evidence of phase transition induced by surface distortion, resistivity is measured as a function of temperature for LSTO films with various thicknesses. However, the transport data of films thinner than 6 u.c. are not shown in the entire temperature range because the resistance becomes too high beyond the measurement capability of our facilities. Figure 2d shows the evolution of resistivity versus temperature curves with varying LSTO thickness. For thicknesses above 6 u.c., the films exhibit metallic characteristics which are similar to the behavior in the bulk $La_xSr_{1-x}TiO_3$[26]. The 6 u.c. thick film exhibits a semiconducting or insulating behavior and the resistivity is an order of magnitude higher than that of the 8 u.c. film at 300 K, clearly indicating the MIT. A detailed analysis of transport properties is described in Supplementary Note 3. It is worth noting that the CO phase usually exhibits an insulating characteristic. To verify the influence of surface distortion, we grew 10 u.c. thick of STO capping layer on the LSTO film of various thicknesses and observed its significant effects on the transport properties (see Supplementary Note 4). Specifically, 6 u.c. thick LSTO films exhibited an insulating behavior without the capping layer but showed metallic behavior with the capping layer of 10 u.c. thickness (Supplementary Fig. 6). These results suggest that the insulating ground state in ultrathin LSTO film originates from the top surface, rather than from the bottom interface or the bulk interior of the film.

To obtain further insight into the thickness-driven MIT, we carried out spectroscopic ellipsometry. The ellipsometry probes photon behaviors scattered by conduction electrons (coherent scattering) and localized electrons (incoherent scattering). Figure 2e shows the real part of the optical conductivity spectra $\sigma_1 (\omega)$ for the LSTO films of various thicknesses $t$. For the 20 u.c. thick LSTO films, whose electronic properties are nearly equivalent to that of the bulk phase, a simple Drude mode (i.e., coherent mode) is clearly observed, as shown in Fig. 2e. As film thickness decreases, the lowest energy region of $\sigma_1 (\omega)$ decreases, while the hump feature at 1.3 eV, indicative of the incoherent mode, increases for thinner films. This crossover from coherent to incoherent modes continues down to $t = 6$ u.c., and at thinner $t$, the Drude mode is abruptly suppressed and only the incoherent mode remains, indicative of the MIT. The existence of the incoherent mode in our spectra suggests the formation of the localized band of the carriers within the CTE gap. It should be noted that the incoherent mode near 1.3 eV in our LSTO film cannot be explained by quasi-classical polarons. The quasi-classical polaron state in doped $SrTiO_3$ is reported to form below 0.5 eV[27,28], which is significantly lower than the energy level observed in our case. However, in ordered phases such as the CO and orbital ordering (OO), the inter-site hopping of electrons is often disrupted by Coulomb repulsion, leading to an increase in the optical gap[29]. Previous reports have indicated that transition metal oxide ultrathin film undergoing a correlated induced MIT exhibit an absorption near 1.5 eV[30,31]. Based on the findings, we speculate that the adsorption near 1.3 eV in ultrathin LSTO film is likely attributed to the correlated induced MIT and is reminiscent of the CO phase. Note that the incoherent mode in our optical conductivity spectra (Fig. 2e) well matched with the energy of the occupied $d$ band is ~1 eV below that of the lowest unoccupied state in metastable CO state at bulk LSTO (Supplementary Fig. 1).

Next, we carry out atomic-scale structural and spectroscopy imaging, Coherent Bragg rods analysis (COBRA), and STEM-EELS to

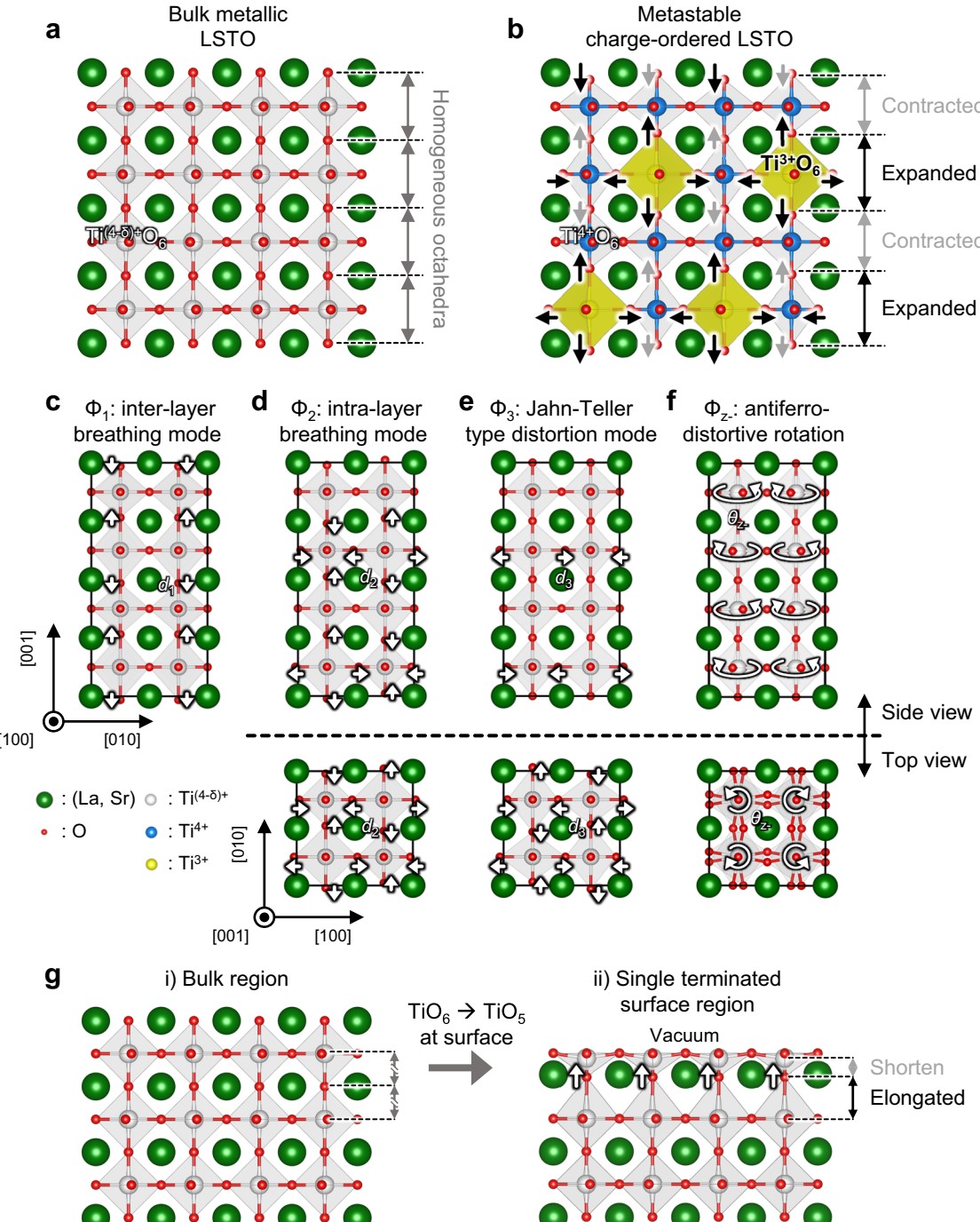

**Fig. 1 | Experimental design for the charge-ordered phase in electron-doped STO. a** Schematics of charge and lattice structure for the metallic phase at bulk LSTO. **b** Schematic drawing of the periodic modulation of lattice distortion and electron density for the metastable insulating charge ordering (CO) phase at the bulk LSTO. Note that the periodicity of lattice distortion is coincident with the modulation of electron density, indicating the CO phase. **c–f** The distortion modes of the CO phase at bulk LSTO: the inter-layer $TiO_6$ breathing mode ($\Phi_1$) (**c**), the intra-layer $TiO_6$ breathing mode ($\Phi_2$) (**d**), the Jahn-Teller type distortion ($\Phi_3$) (**e**), and the antiferrodistortive rotation ($\Phi_{z-}$) (**f**). **g** Schematic of surface distortion which occurs at (001) oriented single terminated surface. Apical oxygen which is located just below the topmost surface is pulled up toward the surface. In (**a–g**) Green, red, blue, yellow, and white spheres represent the (La, Sr), O atoms and $Ti^{4+}$, $Ti^{3+}$, and partially filled $Ti^{(4-\delta)+}$ cations, respectively. In (**b**) the yellow shades on the $Ti^{3+}O_6$ octahedral surface are guides for eyes.

explore atomic and electronic structures in LSTO thin film. Figure 3b displays representative COBRA fits to experimentally measured crystal truncation rods (CTRs) by synchrotron surface X-ray diffraction on LSTO 8 u.c./STO at room temperature. The reconstructed 2D electron density map of LSTO sliced through the (110) atomic plane is shown in Fig. 3a. All the atomic positions, including oxygen atoms, are clearly visible as discrete peaks in the electron density map. Since lattice modulation is reminiscent of the CO phase, we focus on the layer-

resolved oxygen octahedral distortion that is represented by the apical oxygen to oxygen distance ($O_{Api.} - O_{Api.}$ distance), i.e., the height of $TiO_6$ octahedron (Fig. 3c and Supplementary Fig. 10). Remarkably, COBRA results show that the expanded and contracted oxygen octahedra alternate along the **c**-axis, indicating the presence of periodic octahedral distortion in the LSTO film. Note that the two-sublattice modulation of $TiO_6$ octahedron height (Fig. 3c) corresponds to inter-layer breathing mode ($\Phi_1$) (Fig. 1c).

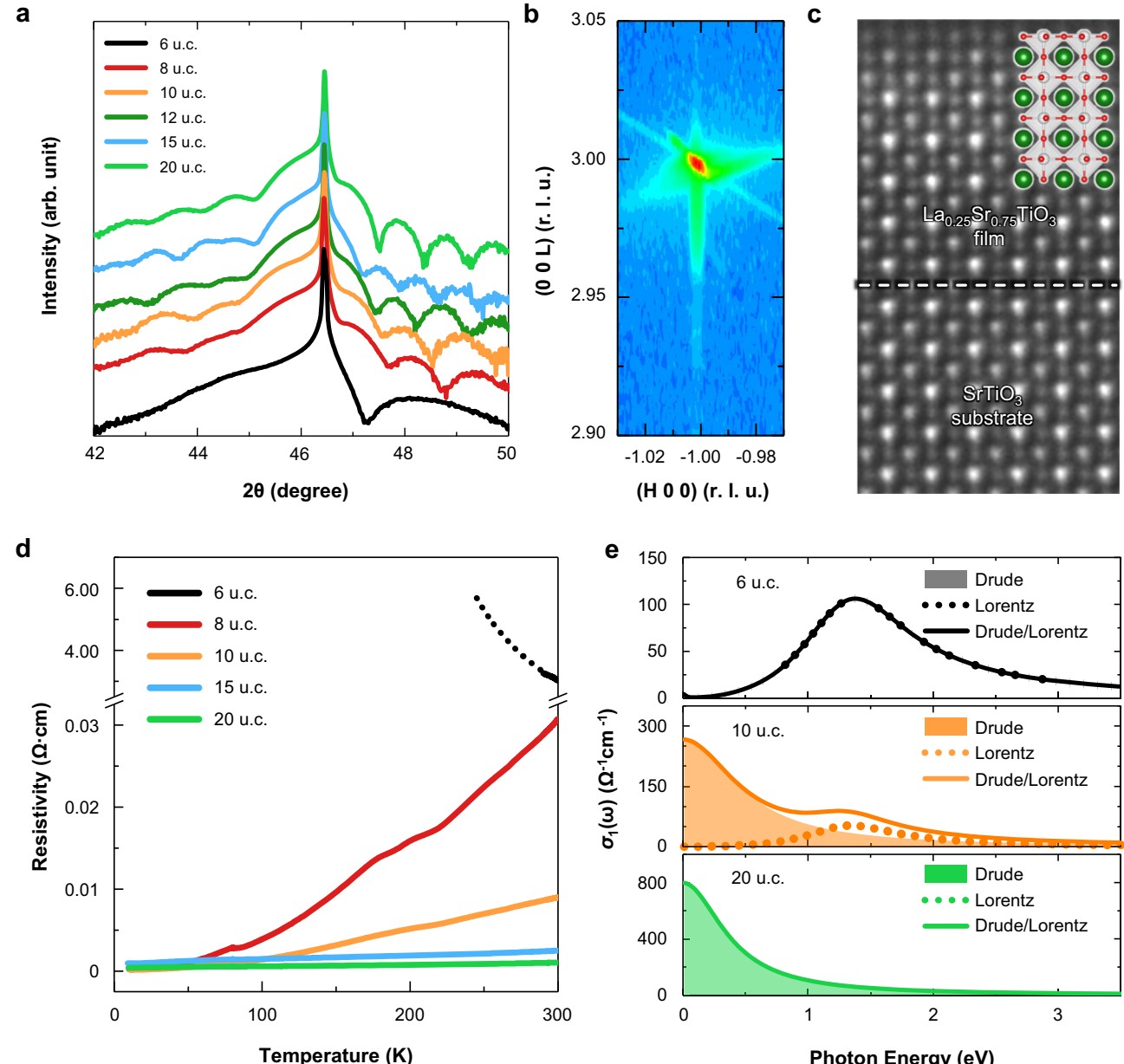

**Fig. 2 | Experimental results of thickness-dependent metal-to-insulator transition. a** A θ-2θ diffraction for the LSTO films, with varying film thicknesses, grown on the STO (001) substrate. **b** Reciprocal space mapping shows X-ray diffraction intensity around (103) Bragg reflections of a 20 unit cells (u.c.) thick LSTO film grown on a STO substrate. **c** Inverse annular bright-field STEM images of a LSTO (6 u.c.)/STO heterostructure. **d** Resistivity of LSTO films with varying film thicknesses as a function of temperature. The black dotted line is an extension of temperature-dependent resistivity curve, following an adiabatic small polaron hopping model (See Supplementary Note 3). **e** Optical conductivity spectra of the LSTO films varying with film thickness. The filled area, dotted, and solid lines are Drude, Lorentz, and Drude/Lorentz peaks, respectively. For clarity, the high energy of charge transfer excitation was removed from the optical conductivity spectra. Source data are provided as a Source Data file.

The electron density is also examined by STEM-EELS analysis. Since the electronic structures of transition metal oxides strongly depend on the filling of their $d$-orbital state, normalized layer-resolved Ti-$L_{2,3}$ edge spectra are collected through the entire thickness of the LSTO films (Supplementary Fig. 11). The variation of the Ti valence could be estimated from the fine details in the Ti $L_{2,3}$ edge[32]. As shown in Fig. 3d, we quantitatively calculate the $Ti^{3+}$ fraction using the standard spectra of STO ($Ti^{4+}$) and LTO ($Ti^{3+}$)[32,33]. The proportion of $Ti^{3+}$ clearly oscillates and diminishes near the STO substrate region, indicating the periodic electron density modulation in the LSTO film. Taken together, our observations from COBRA and EELS analysis reveal strong evidence of the periodic charge and structural modulation. Starting from the -1st layer (i.e., the subsequent layer of the topmost surface), the longer apical oxygen to oxygen distance (Fig. 3c) is first observed with localized electrons (i.e., $Ti^{3+}$ state) (Fig. 3d). The -2nd layer has the opposite behavior, i.e., shorter apical oxygen to oxygen distance (Fig. 3c) with the formation of the $Ti^{4+}$ state (Fig. 3d). Interestingly, this phase penetrates 6 u.c. from the surface, forming periodic modulation of the electron density and lattice distortion which eventually disappears near the interface between the LSTO film and substrate. It should be noted that the 6 u.c. is the same as the critical thickness for the MIT in LSTO.

To understand the underlying physics of the above system, we investigate the atomic and electronic structures of the LSTO films with various thicknesses $t$, on the STO substrate (see Supplementary Note 8) based on first-principles calculations. In a nutshell, the

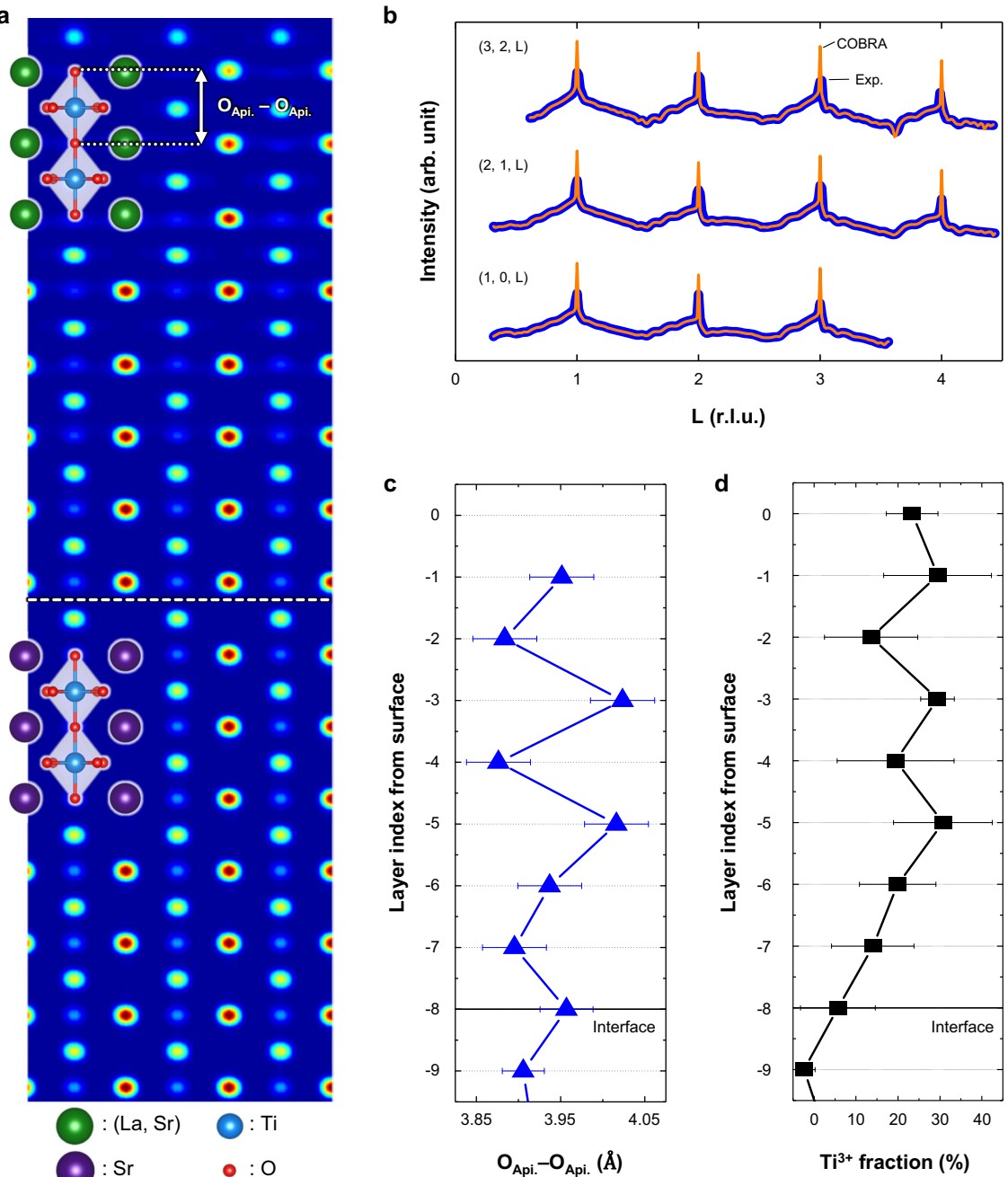

**Fig. 3 | Experimental evidence for the periodic modulation of oxygen octahedral distortion and electron density. a** A 2D vertical slice through the 3D electron density map along the (110) atomic plane for an 8 unit cells (u.c.) thick LSTO film. Equatorial oxygen is not visible along the (110) atomic plane. Green, purple, blue, and red spheres represent the (La, Sr), Sr, Ti, and O atoms, respectively. **b** Representative measured crystal truncation rods, labeled as (H, K, L), where the intensity versus the reciprocal vector L is shown (blue dots) for LSTO 8 u.c./STO at room temperature, as well as the corresponding COBRA fits (orange solid lines). **c** A layer-resolved apical oxygen-to-oxygen distance (i.e., the height of $TiO_6$ octahedron), which is extracted from COBRA. The experimental errors are estimated by using the COBRA noise analysis method. **d** STEM-EELS results which clearly show electron density modulation in alternating layers. The error bars show the standard errors of consecutive scans. Source data are provided as a Source Data file.

calculation results show the metal ($t > 6$ u.c.) to insulator ($t \leq 6$ u.c.) transition (Fig. 4a and Supplementary Fig. 13) with the periodic modulation of the lattice and the electron density, i.e., alternating $Ti^{3+/4+}O_6$ and $Ti^{4+}O_6$ octahedral layers, penetrating up to 6 u.c. from the surface into the interior regardless of the film thickness, which is fully consistent with the experimental observation (Fig. 2c, d). For an ultrathin film ($t \leq 6$ u.c.), the entire film becomes the modulated structure that contains alternating layers with expanded and contracted $TiO_6$ octahedra. Figure 4b–d shows the obtained DFT results of the 6 u.c. thick

LSTO film ($t = 6$ u.c.) on STO. The lattice modulation with $Ti^{3+/4+}O_6$ and $Ti^{4+}O_6$ octahedra is represented by schematics (Fig. 4b). The layer-averaged height of $TiO_6$ octahedron (i.e., $O_{Api.} - O_{Api.}$ distance), and the layer-averaged volume of $TiO_6$ (Fig. 4c) are quantitatively analyzed to compare with our experimental result (Fig. 3c). The calculated layer-resolved density of states (DOS) shows the periodic modulation of the electron density (Fig. 4d).

We further analyze the obtained atomic and electronic structure in detail, and the lattice modulation by decomposing it into the 4

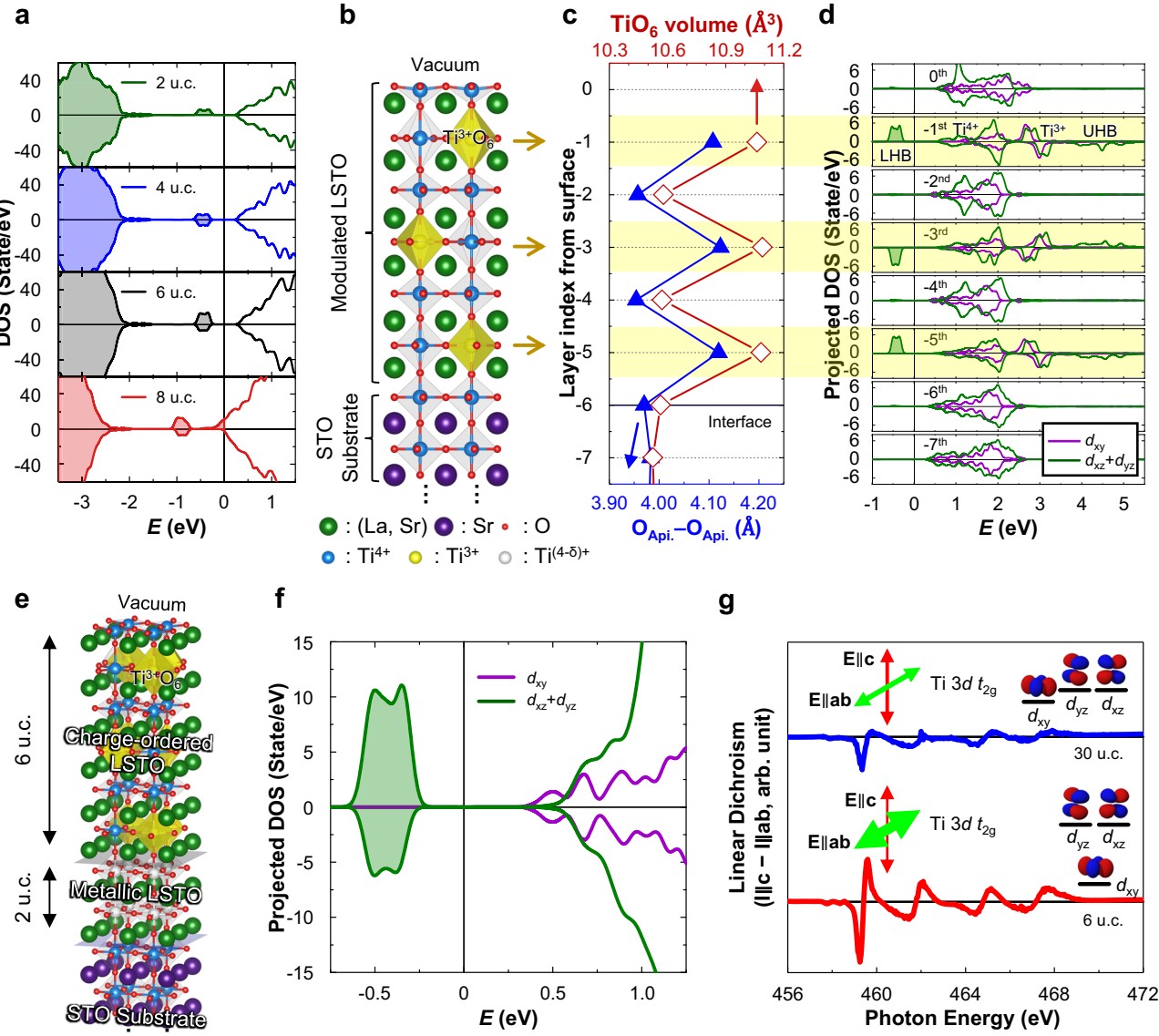

**Fig. 4 | Atomic and electronic structure of LSTO film on STO (001). a** Calculated spin-polarized density of states (DOS) of the LSTO/STO slab with various film thicknesses of LSTO $t$ = 2, 4, 6, and 8 unit cells (u.c.). The spurious electronic states originating from the fixed bottom layer of the slab are excluded from the DOS (see Supplementary Note 8). For $t$ = 2, 4, and 6 u.c., the midpoint of the energies of the lowest unoccupied state and the highest occupied state is set to be zero and represented by the vertical line. For $t$ = 8 u.c., the Fermi energy is set to be zero and represented by the vertical line. **b–d** Calculated atomic and electronic structures of the LSTO/STO slab with $t$ = 6 u.c. **b** Schematic drawing of the atomic structure of the slab. **c** The layer-averaged height (blue line with the closed triangles) and volume (red line with the open diamonds) of TiO$_6$ octahedra of the slab. **d** The layer-resolved DOS projected onto the Ti $3d$ $t_{2g}$ orbitals. The lower Hubbard bands (LHB)

and the upper Hubbard bands (UHB) of the odd-numbered sublayers are at $E$ ~ −0.5 and 4.5 eV, respectively. **e** Schematic representation of the atomic structure of the LSTO film on STO (001) surface for $t$ = 8 u.c. **f** Calculated DOS of the charge-ordered LSTO film ($t$ = 6 u.c.) projected onto the Ti $3d$ $t_{2g}$ orbitals. **g** X-ray linear dichroism (XLD) spectra around the Ti-$L_{2,3}$ edge of the LSTO film for $t$ = 30 u.c. (blue line), and 6 u.c. (red line). In (**a**, **d**, **f**) positive and negative regions correspond to the up-spin and the down-spin states, respectively. In (**b**, **e**) green, purple, red, blue, yellow, and white spheres represent the (La, Sr), Sr, O atoms and Ti$^{4+}$, Ti$^{3+}$, and partially filled Ti$^{(4-\delta)+}$ cations, respectively. In (**d**, **f**) purple and green lines represent the DOS projected onto the Ti $d_{xy}$ and $d_{xz}$ + $d_{yz}$, respectively. The midpoint of the energies of the lowest unoccupied state and the highest occupied state is set to be zero and represented by the vertical line. Source data are provided as a Source Data file.

representative octahedral distortion modes: the inter-layer breathing mode ($\Phi_1$), intra-layer breathing mode ($\Phi_2$), Jahn-Teller type distortion mode ($\Phi_3$), and the antiferrodistortive rotation ($\Phi_{z-}$) (Supplementary Fig. 14). Note that the analyzed results are consistent with metastable CO state in bulk LSTO (see Fig. 1c–f, and Supplementary Note 1). Using the obtained distortion modes and the calculated layer-resolved DOS, we explain the atomic and electronic structures of the film and their coupling layer-by-layer. On the -1st, -3rd, and -5th layers (odd-numbered sublayers), TiO$_6$ octahedra expand in the out-of-plane direction ($\Phi_1$) (Fig. 4c and Supplementary Fig. 14a) and some of them expand more than the others ($\Phi_2$) (Supplementary Fig. 14b). Consequently, the

out-of-plane $t_{2g}$ orbitals ($d_{xz}$/$d_{yz}$ orbitals) of Ti ions in the more expanded octahedra become localized and energetically stable (Supplementary Fig. 16), -1 eV below than the other $t_{2g}$ states (Fig. 4d). In contrast, the −2nd and -4th layers' TiO$_6$ octahedra slightly contract in the out-of-plane direction (Fig. 4c and Supplementary Fig. 14a), and the energy levels of the $t_{2g}$ states are relatively unchanged except for the slight increase of the $d_{xz}$/$d_{yz}$ band energy (Fig. 4d). As a result, all the electrons from the La$_{Sr}$ dopants occupy the low-lying localized $d_{xz}$/$d_{yz}$ band of the odd-numbered sublayers opening the Hubbard gap, while all the $t_{2g}$ states of the even-numbered sublayers are unoccupied (Fig. 4d). Hence, the entire LSTO film becomes an insulator which

consists of alternating layers of Mott insulator (odd-numbered sublayers) and band insulator (even-numbered sublayers) as shown in Fig. 4d. The 2 and 4 u.c. thick LSTO film ($t$ = 2, 4 u.c.) showed similar results to that of 6 u.c. thick film (Supplementary Fig. 17).

For LSTO films thicker than the characteristic length ($t$ > 6 u.c.), the top 6 layers with the CO state are sitting on top of the metallic layers which do not have such a distortion. Figure 4e represents the DFT results of the LSTO film with $t$ = 8 u.c. on STO. Indeed, the top 6 layers of the film show the periodic distortion with the expanded and contracted Ti octahedra, resulting in alternating layers with the electron density modulation (Supplementary Fig. 18) similar to the $t$ = 6 u.c. (Fig. 4d). The -6th and -7th layers remain a metallic state with the occupied conduction bands owing to the absence of the periodic distortion (Supplementary Fig. 18). To summarize, the periodic distortion penetrates the film up to 6 u.c., and the atomic and electronic structures are nearly identical regardless of $t$. Thus, when $t$ is below the penetration length ($t \leq 6$ u.c.), the entire film becomes insulating CO states (Fig. 4a, d and Supplementary Fig. 17), while when $t$ is larger than the penetration length ($t$ > 6 u.c.), the metallic region remains in the film (Fig. 4a and Supplementary Fig. 18).

To further support validity of DFT predicted electronic structures for the charge-ordered phase in LSTO film, we performed synchrotron-based X-ray absorption spectroscopic (XAS) since XAS can directly probe the unoccupied states (i.e., conduction band) near the Fermi energy with chemical and orbital sensitivity[34–36]. As shown in Fig. 4f, DOS of the charge-ordered LSTO film ($t$ = 6 u.c.) obtained from DFT calculation shows predominant Ti $d_{xy}$ orbital polarization in the lowest unoccupied state (i.e., conduction band edge). This originates from the contracted $TiO_6$ oxygen octahedra in the [001] direction within the −2nd and -4th layers (Fig. 4b–d) in the inter-layer breathing mode ($\Phi_1$). We collected XAS spectra at the Ti-$L_{2,3}$ absorption edge with photon polarization **E** ∥ **ab** (parallel to the sample plane) and **E** ∥ **c** (almost perpendicular to it) in total electron yield mode. The difference between the spectra of the two polarization configurations is pronounced when $t$ = 6 u.c., while the difference is smaller for thicker films ($t$ = 30 u.c.) (Fig. 4g). In the case of thinner films (i.e., charge-ordered insulating film), in-plane orbital ($d_{xy}$) is found to be lowest unoccupied state of conduction band since the energy of $L_3$:$t_{2g}$ peak obtained by **E** ∥ **c** is much higher than that obtained by **E** ∥ **ab** (Fig. 4g). These results reveal that the lowest unoccupied state of the ultrathin LSTO film consists of $d_{xy}$ orbital, which is consistent with those obtained from the DFT calculations of the charge-ordered phase in LSTO film (Fig. 4f).

The LSTO surface structure in our case stabilizes a charge-ordered Mott insulating phase with a nearly half-filled Ti 3$d$ state of odd-numbered sublayers, which is far below the known onset of the electron concentration for the Mott transition, i.e., La ~95%[26]. This is because of the orbital selective Mott transition (OSMT) mechanism, i.e., the system consisting of lower-lying $d_{xz}$/$d_{yz}$ bands reduces the critical interaction strength for Mott transition through the electron localization due to the symmetry breaking[37–39]. It is also worth noting that stripe-like unidirectional CO has been reported in SrO-terminated STO surface[40]. This takes a place only at the reconstructed surface regime, implying periodic electron and lattice modulations exist along the plane. It should be noted that we could not find any evidence of the reconstructed structure on the surface of the LSTO thin film (see Supplementary Note 9). Another possible mechanism to explain surface-driven MIT is the electron draining originated from oxygen adsorbates on the film surface[41]. However, it has been reported that such an effect is limited to 1 or 2 u.c. of the topmost surface[41]. Considering the relatively thicker critical thickness for the MIT in LSTO thin film than the thickness of an electrical dead layer originated from oxygen adsorbates[41], we can exclude the effect of oxygen adsorbates as the origin of the thickness-driven MIT in the LSTO film (Supplementary Note 10).

In the present work, a surface dangling bond is key to triggering the emergence of the CO insulating phase. We first confirm that surface distortion, which originates from the dangling bond in the perovskite oxide thin film[42], through surface X-ray diffraction measurements and performed COBRA analysis (see Supplementary Note 9). The slices along the (200) and (110) atomic planes reveal a downward rumpling on the 0$^{th}$ $TiO_2$ sublayer (top surface) and upward rumpling on the -0.5th (La, Sr)O sublayer (Supplementary Fig. 20). Note that experimentally determined surface structures such as atomic rumpling behavior at the surface layer is in a good agreement with those obtained from the DFT calculations. Due to the dangling bond of Ti ions on the surface, i.e., the decreased coordination number of Ti atoms on the top surface layer, the apical oxygen of the octahedron just below the surface is pulled up towards the surface through Pauling's second rule[43,44] (Fig. 1g). With this constrained, near the surface, metastable CO phase becomes more stable than the metallic phase through the activation of the inter-layer breathing mode ($\Phi_1$) (Figs. 1c, 3a and Supplementary Fig. 14a). In addition, surface distortion is attenuated at depth due to the energy competition between the metastable CO insulating phase and the bulk-like metallic phase, which is schematically shown in Supplementary Fig. 21. Calculating quantitative values such as energy difference and the transition barrier energy between the metallic phase and the metastable CO phase of the bulk would be interesting and helpful to understand the damping behavior. But it's beyond the scope of this work. In addition, we anticipate that the CO phase is not limited to a specific doping concentration such as 25% $La_{Sr}$, but could potentially exist in a certain range of doping concentrations in the $La_xSr_{1-x}TiO_3$ system (see Supplementary Note 12).

The stability of the CO phase in the thin film of electron-doped STO originates from the nonequilibrium inter-layer breathing mode along the surface normal direction, induced by surface distortion occurring in the (001) oriented thin film. Our findings establish a framework for understanding phase transition phenomena in oxide heterostructures as the thickness is reduced below a certain critical thickness[30,31,35,45–47]. We envision that our approach to stabilizing non-equilibrium states will serve as a fertile platform for engineering CO phase related functional properties, such as superconductivity, colossal magnetoresistance, and multiferroicity in complex oxides.

## Methods

### Sample fabrication and electrical characterization

LSTO thin films were epitaxially grown on $TiO_2$-terminated $SrTiO_3$ (001) substrates using pulsed laser deposition (PLD) with in-situ reflection high energy electron diffraction monitoring. To obtain the $TiO_2$-terminated substrates, as-received STO substrates were etched using buffered-HF for 45 s and annealed at 1100 °C for 6 h. Before the film growth, we performed the second chemical etching procedure using buffered-HF for 10 s, to stabilize $TiO_2$-terminated surface of the STO substrate[48]. During the film growths, the temperature of PLD sample heater was kept at 650 °C. The chamber pressure was $10^{-5}$ Torr consisting of a mixture of oxygen/ozone. The surface morphology was examined using atomic force microscopy, and the surface roughness was below 1 u.c. for all samples. The electrical transport was measured using a physical-property measurement system (PPMS) under a rough vacuum of a $10^{-3}$ Torr. Au/Pt were used as contact metals for van der Pauw geometry.

### Density functional theory calculations

The first-principles density functional theory (DFT) calculations were performed with the Vienna ab initio simulation package (VASP)[49,50]. The projected augmented wave (PAW) method[51] with the 520 eV of a kinetic energy cutoff and spin-polarized generalized gradient approximation (GGA) with Perdew-Burke-Ernzerhof (PBE)[52] scheme were used. Spin-polarized calculation and the semi-empirical Hubbard

+U approach[53,54] with rotationally invariant formalism by Liechtenstein[55] were used to better description for exchange and correlation effects. Here we used $(U − J) = 5 − 0.64$ eV for Ti 3d orbitals[56–59], while $(U − J) = 11 − 0.68$ eV for La 4f orbital to avoid spurious mixing with the conduction band of STO[57,60,61]. For lanthanum, strontium, titanium, and oxygen, the valence electron configurations of $5s^2 5p^6 6s^2 4f^1$, $4s^2 4p^6 5s^2$, $3s^2 3p^6 4s^2 3d^2$, and $2s^2 2p^4$ were considered, respectively. The Hellman–Feynman force convergence limit of 0.01 eV/Å is used for the atomic structure optimization. For the LSTO bulk calculation, Γ-centered $8 × 8 × 4$ k-point meshes are used for the atomic structure optimization and the self-consistent calculations for the charge-convergence. For the slab calculation, Γ-centered $4 × 4 × 1$ and $6 × 6 × 1$ k-point meshes with 0.05 eV of Gaussian smearing are used for the atomic structure optimization and the self-consistent calculations for the charge-convergence, respectively. The details can be found in Supplementary Notes 1, 8. 3D structural visualization program VESTA was used to show the atomic structure[62].

## STEM and EELS measurements

The specimens for scanning transmission electron microscopy (STEM) were prepared by using a triple-beam focused ion beam system (NX2000, Hitach). The specimens were thinned by using the $Ga^+$ ion beam for electron beam transparency with decreasing the acceleration voltage from 30, 5 to 1 kV. Subsequently, the $Ar^+$ ion beam at 1 kV was used for removing the surface damages on both sides of the specimen. STEM imaging and electron energy loss spectroscopy (EELS) were carried out on a STEM-equipped Cs corrector (JEM-ARM 300 F, JEOL). The convergence angle of the electron probe for STEM was 34.8 mrad. The EELS data was obtained by using an energy filter (Gatan GIF Quantum ER 965, USA). The energy resolution was estimated as 0.6 eV from the full width at half maximum of zero loss peak. The $Ti^{3+}$ fraction was calculated by using multiple linear least square (MLLS) fitting on the Ti-$L_{2,3}$ edge. The reference EEL spectra of $Ti^{4+}$ and $Ti^{3+}$ were obtained from single crystal $SrTiO_3$ and $LaTiO_3$ thin films, respectively. The reference EEL spectra were used for the MLLS method to quantitatively evaluate the $Ti^{3+}$ fraction from the EELS Ti-$L_{2,3}$ data.

## Synchrotron crystal truncation rod and coherent Bragg rod analysis

Synchrotron X-ray surface diffraction crystal truncation rod (CTR) measurements were carried out to quantify the atomic structures with a sub-Å precision of STO (8 u.c.) grown on STO (001) substrate. The CTR measurements were performed on a Newport Kappa six-circle diffractometer using a X-ray energy of 20 KeV at sector 33-ID-D of the Advanced Photon Source, Argonne National Laboratory. The total X-ray flux is about the $2.0 × 10^{12}$ photons s$^{-1}$. The X-ray beam was focused by a pair of Kirkpatrick–Baez mirrors down to a beam profile of ~80 μm (vertical) × 200 μm (horizontal). The two-dimensional diffraction images of CTRs at the out-of-plane **L** direction in the reciprocal space were taken by a pixel array area detector (Dectris PILATUS 100 K). Samples were protected under dry helium gas flow in a concealed sample cell during room temperature measurements. Three-dimensional total electron densities for the complete atomic structures of the thin film system (e.g., epitaxial thin film unit cells and top few unit cells of the substrate) were reconstructed from the complete set of CTRs by using an iterative phase retrieval technique, known as coherent Bragg rods analysis (COBRA)[63].

## X-ray absorption and X-ray linear dichroism (XLD) measurements

The XAS and XLD measurement has been performed at the 2 A beamline of the Pohang Accelerator Laboratory at 300 K, in total electron yield mode. The spectra were measured at the Ti-$L_{2,3}$ edge for the two polarizations (in-plane and out-of-plane). The direction of the soft X-ray polarization vector **E** was changed by 90° to obtain the in-plane ($E \parallel ab$) and the out-of-plane ($E \parallel c$) orbital responses, while the angle of incidence was 70° with respect to the surface normal direction. Soft X-rays with their electric-field orientation perpendicular to the **c**-axis ($E \parallel ab$) are more sensitive in probing the state with $d_{xy}$ orbitals. The X-ray linear dichroism (XLD) signals were derived from the intensity difference ($I \parallel c − I \parallel ab$) of normalized XAS spectra.

## Optical spectroscopy

The optical conductivity spectra were obtained by using the combined spectroscopic techniques, i.e., spectroscopic ellipsometry and reflectivity measurement. The spectroscopic ellipsometry with variable incident light angles (60°, 70°, and 75°) was performed in the range of 0.7 eV to 6 eV at room temperature. The measured ellipsometry angular spectra $\Psi(\omega)$ and $\Delta(\omega)$ were fitted with the two-layer model (LSTO film + STO substrate) to obtain the complex optical conductivity spectra, $\tilde{\sigma}(\omega) [\equiv \sigma_1(\omega) + i\sigma_2(\omega)]$. The fitting spectra are composed of Drude and tauc-Lorentz modes. The optical data were also confirmed by comparing a reflectivity directly measured by the conventional FT-IR/grating-type spectrometers in a photon energy range of 0.1 eV–6 eV.

## Data availability

The data that support the findings of this study are available from the corresponding author upon request. Source data are provided with this paper.

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

## Acknowledgements

This work was supported by the National Research Foundation of Korea through the Basic Science Research Program (NRF–2022R1A2C200 4868) and KISTI Supercomputing Center (KSC–2019-CRE-0113). This research is funded by the Gordon and Betty Moore Foundation's EPiQS Initiative, grant GBMF9065 to C.B.E., Vannevar Bush Faculty Fellowship (N00014–20-1–2844 (C.B.E.)). Transport measurement at the University of Wisconsin–Madison was supported by the US Department of Energy (DOE), Office of Science, Office of Basic Energy Sciences (BES), under award number DE-FG02-06ER46327. The authors acknowledge partial support of this research by NSF through the University of Wisconsin Materials Research Science and Engineering Center (DMR–2309000). K.E. would like to acknowledge the support by National Research Foundation of Korea through the Basic Research Program (NRF-2022R1C1C2010693). S.O. would like to acknowledge the support by National Research Foundation of Korea through the Basic Research Program (NRF-2021R1I1A1A01058779). I.S. and Y.S.L. would like to acknowledge the support by Basic Science Research Program through the National Research Foundation of Korea (NRF) funded by the Ministry of Education (2021R1A6A1A10044154). This research used resources of the Advanced Photon Source, a U.S. Department of Energy (DOE) Office of Science User Facility, operated for the DOE Office of Science by Argonne National Laboratory under Contract No. DE-AC02-06CH11357. The STEM work by J.S. and S.H.O. was supported mainly by the Samsung Research Funding & Incubation Center of Samsung Electronics under Project Number SRFC-MA1702-01 and partly by the National Research Foundation of Korea (NRF) funded by the Korea government (MSIT) (No. NRF-2020R1A2C2101735), Creative Materials Discovery Program (NRF-2019M3D1A1078296), the KENTECH Research Grant (KRG2022-01-019). J.J. and S.Y.C. acknowledge the support by Korea Basic Science Institute (National research Facilities and Equipment Center) grant funded by the Ministry of Education (2020R1A6C101A202) and National R&D Program through the National Research Foundation of Korea (NRF) funded by Ministry of Science and ICT (RS-2023-00258227). H. L. acknowledges the support by National Research Foundation of Korea (NRF) grant funded by the Korea government (MSIT) (No. 2021R1A4A1032085).

## Author contributions

K. E. and J.L. conceived the project. K. E. and K. L. carried out the thin film growth. B. C. and S. O. performed density functional theory calculations. K. E. and J.W.L. performed the XRD work. H. Z. performed COBRA measurement and data analysis. J. J., S.-Y. C., J. S., and S. H. O. performed scanning transmission electron microscopy measurement. K. E., M. C., Y. K, and H. L. performed transport measurement. I. S. and Y. S. L. performed ellipsometric spectroscopy. K. E., B. C., S. O., M. S. R., C.B.E., and J.L. prepared the manuscript. J.L. and C.B.E supervised the study and co-wrote the paper. All authors discussed the results and commented on the manuscript.

## Competing interests

The authors declare no competing interests.
