## [Peer Review File · Nature Communications]

REVIEWER COMMENTS

Reviewer #1 (Remarks to the Author):

In this work, the team led by Profs. Lee and Eom reported the design principle and observation of charge order state, a fundamental order state in strongly correlated system, in electron doped SrTiO₃ thin films. This order state is stabilized by the surface induced structural deformation, which lead to alternative stacking of different octahedral layers (that is the elongated and shorten bonds). Experimentally, the work is very solid with several state-of-art techniques employed to support their claim of charge order state. The paper was also nicely written up with very smooth logic flow. I believe that it could be an attractive work for a broad audience, finding more broad applications in other strongly correlated oxides. Therefore, I would suggest the publication of the current work in Nature Communications with the following suggestions, which I believe would be able to further strength the scientific merits of the current work.

- 1) The authors claimed that the charge order phase is stabilized by the surface state. Then can the authors show the result of 6 uc LSTO grown on STO with a thin layer STO (>10 uc) as capping. This study can testify the influence of the surface termination.
- 2) In this work, the authors observed the A-type charge ordering due to the doping level of 25% La. Can the authors repeat the study (6 uc) with other doping level, say 15%, which is not the magic number.
- 3) What the activation energy from the transport data? Whether this is the same as the optical band gap? Also, the 8 uc sample is also very exotic with extremely large RRR. Can authors comment on this?
- 4) It is not clear how the authors modeled the surface state during their calculations.

Reviewer #2 (Remarks to the Author):

The paper reports the dead layer formation in the La-doped SrTiO₃ (LSTO) thin film deposited on a SrTiO₃ substrate. The high electrical resistance observed in the film below the critical thickness is attributed to charge ordering. From the COBRA and STEM-EELS measurements, the authors argue on the structures of the heterostructures.

I do not recommend publishing in Nature Communications because the study seems incomplete.

If the surface triggers the metal-insulator transition, it is essential to determine the surface structure experimentally. In particular, it is critical to clarify whether the surface is reconstructed. Investigating the in-plane structures at the atomic level by RHEED, STM, or other methods is recommended. In addition, the discussion is weak because the electronic states are discussed only in terms of electron transport properties. Probing the electronic states near the Fermi level directly using photoelectron spectroscopy is desired to connect with calculation results.

minor issue: Some of the paragraphs are too long to read.

Finally, information should be added on whether the transport property measurements were conducted in non-exposed atmospheres. It is known that oxygen adsorbs on the film surface in ambient air and induces metal-to-insulator transition. Such an effect is also possible.

Response on Reviewers' comments

We thank the Reviewers and the Editor for in-depth reviews and excellent questions/suggestions regarding our manuscript. In the pages that follow, we provide our responses to each of their questions, in order. Original reviewer comments are written in black. The responses are written in blue. Relevant passages from the modified manuscript appear in red.

Reviewer #1 (Remarks to the Author)

- Comments:

In this work, the team led by Profs. Lee and Eom reported the design principle and observation of charge order state, a fundamental order state in strongly correlated system, in electron doped SrTiO₃ thin films. This order state is stabilized by the surface induced structural deformation, which lead to alternative stacking of different octahedral layers (that is the elongated and shorten bonds). Experimentally, the work is very solid with several state-of-art techniques employed to support their claim of charge order state. The paper was also nicely written up with very smooth logic flow. I believe that it could be an attractive work for a broad audience, finding more broad applications in other strongly correlated oxides. Therefore, I would suggest the publication of the current work in Nature Communications with the following suggestions, which I believe would be able to further strength the scientific merits of the current work.

We are glad that the reviewer finds our manuscript “nicely written up with very smooth logic flow” and supports the publication in Nature Communications. The manuscript has been revised to thoroughly address all the reviewer’s comments as follows.

(1) The authors claimed that the charge order phase is stabilized by the surface state. Then can the authors show the result of 6 uc LSTO grown on STO with a thin layer STO (>10 uc) as capping. This study can testify the influence of the surface termination.

As suggested by the reviewer, we grew a 10 u.c. thick STO capping layer on the LSTO thin films of various thicknesses and observed its significant effects on transport properties.

To assess the impact of the capping layer, we measured the room temperature resistivity of the samples with and without the STO capping layer (Fig. R1a). For thinner LSTO films, the resistivity significantly decreased with the growth of the STO capping layer on the top. This effect was less pronounced for thicker films. Particularly, 6 u.c. and 7 u.c. thick LSTO films exhibited an insulating behavior without capping layer but demonstrated metallic behavior with the capping layer of 10 u.c. thickness ((Fig. R1b). These findings suggest that the insulating ground state in ultrathin LSTO films originates from the top surface, as described in the manuscript, rather than from the bottom interface or the bulk interior of the film.

Fig. R1 | Effect of STO capping layer on transport properties of the LSTO films. a, Thickness-dependent resistivity of LSTO films with and without 10 u.c. thick STO capping layer measured at room temperature. **b,** Resistivity as a function of temperature for 6 u.c. and 7 u.c. thick of LSTO films with and without 10 u.c. thick of STO capping layer, respectively.

We added discussions on this issue and additional experimental results into the revised manuscript and supporting information as below.

Main Text:

(Page 6) “To verify the influence of surface distortion, we grew a 10 u.c. thick STO capping layer on the LSTO film of various thicknesses and observed its significant effects on the transport properties (see Supplementary Note 4). Specifically, 6 u.c. thick LSTO films exhibited an insulating behavior without the capping layer but showed metallic behavior with the capping layer of 10 u.c. thickness (Supplementary Fig. 6). These results suggest that the insulating ground state in ultrathin LSTO film originates from the top surface, rather than from the bottom interface or the bulk interior of the film.”

Supporting Information:

“Supplementary Note 4: Effect of the surface distortion on the transport properties” is added in the revised Supporting Information.

(Note 4) Fig. R1 is added in the revised Supporting Information as Supplementary Fig. 6.

(2) In this work, the authors observed the A-type charge ordering due to the doping level of 25% La. Can the authors repeat the study (6 uc) with other doping level, say 15%, which is not the magic number.

We thank the reviewer for the insightful comments. As suggested by the reviewer, we extended our investigation including experimental and DFT approach with varying doping levels such as 15% La_{Sr}, and observed a thickness-driven metal-to-insulator transition (MIT) as well. Our DFT calculations indicate that the insulating ground state in ultrathin film at this doping concentration also originates from the emergence of the CO phase. We attribute that the CO phase is not limited to specific doping concentrations but can exist within a certain range of doping concentrations in La_xSr_{1-x}TiO₃ system. We believe that this subject will be an intriguing area for the future research.

We measured the thickness dependence of the carrier densities at the 15% and 25% La-doped STO thin films (Fig. R2). The sheet carrier density (n_s) versus thickness for both La-doping concentrations shows a linear relationship and each linear fit intersects to $n_s = 0$ at nonzero thickness (Fig. R2a). Below these thicknesses, the resistivity exceeds the measurement limit, indicating thickness-driven MIT. The temperature-dependent resistivity of the 15% La-doped film with varying thicknesses also shows consistent results, implying thickness-driven MIT (Fig. R2b).

Fig. R2 | **a**, Sheet carrier density of 25% and 15% La-doped STO films as a function of the thickness measured at room temperature. The extrapolated dotted lines are linear fits to each plot. **b**, Resistivity of 15% La-doped STO films with varying film thicknesses as a function of temperature range of 80 K to 300 K.

To understand the underlying physics of the thickness-driven MIT at different doping concentrations such as 15% La_{Sr}, we investigated the atomic and electronic structures of the 12.5% La-doped STO film with 6 u.c. thickness on the STO substrate by first-principles calculations. To simulate atomic and electronic structures of 15% La-doped STO film on the top of the STO (001) using a conventional supercell approach, a supercell size should be

increased by a minimum of $2\sqrt{2}a \times 5\sqrt{2}a$ along the in-plane direction, which has 20 Sr sites on every SrO sublayer. Considering computational resources, such a supercell approach is limited so calculation convergences cannot be achieved. Therefore, we first consider using virtual crystal approximation (VCA) [*Phys. Rev. B* **61**, 7877 (2000)] to simulate 15% La-doped STO film. Using VCA, we construct 15% La-doped STO film by replacing the La and Sr atoms' pseudopotentials with a weighted pseudopotential by La/Sr ratio 0.15/0.85, thus the size of the supercell can be effectively reduced. However, when the local potentials of the two atoms are not sufficiently comparable, the result of VCA is usually unreliable [*Phys. Rev. B* **89**, 165201 (2014)]. Hence, we decided to construct the 12.5% La-doped STO film, which slightly deviates from 15% La_{Sr} , using the supercell approach. In the in-plane directions, the $2\sqrt{2}a \times 2\sqrt{2}a$ cell, which has eight Sr sites on every SrO sublayer, is used to simulate the 12.5% La-doped STO.

The calculation results show an insulating phase with the periodic modulation of the lattice and the electron density, i.e., alternating $\text{Ti}^{3+/4+}\text{O}_6$ and Ti^{4+}O_6 octahedral layers. Figure R3 shows the obtained DFT results of the 12.5% La-doped STO film with 6 u.c. thickness on STO. The lattice modulation with $\text{Ti}^{3+/4+}\text{O}_6$ and Ti^{4+}O_6 octahedra is represented by schematics (Fig. R3a). The layer-averaged height of TiO_6 octahedron (i.e., $O_{\text{Api}} - O_{\text{Api}}$ distance), and the layer-averaged volume of TiO_6 (Fig. R3b) are quantitatively analyzed, showing a periodic modulation of TiO_6 octahedron. The calculated layer-resolved density of states (DOS) shows the periodic modulation of the electron density (Fig. R3c).

Fig. R3 | Atomic and electronic structures of the 12.5% La-doped STO/STO slab with 6 u.c. thickness. **a**, Schematic representation of the atomic structure of the 12.5% La-doped STO film on STO (001) surface along [110] direction. Green, purple, red, blue, and yellow spheres represent the (La, Sr), Sr, O atoms and Ti^{4+} , and Ti^{3+} cations, respectively. The yellow shades on the Ti^{3+}O_6 octahedral surface are guides for eyes. **b**, The height and volume of TiO_6

octahedra of the LSTO/STO heterostructures. The layer-averaged height and volume of the TiO_6 octahedra obtained from the DFT calculations are plotted. The blue line with the closed triangles and the red line with the open diamonds represent the layer-averaged height and volume of TiO_6 octahedra, respectively. **c**, The layer-resolved DOS projected onto the Ti $3d t_{2g}$ orbitals. Purple and green lines represent the DOS projected onto the Ti d_{xy} and $d_{xz}+d_{yz}$, respectively. Positive and negative regions correspond to the up-spin and the down-spin states, respectively. The Fermi energy is set to be zero and represented by the vertical line. In **b,c**, the yellow highlights for the -1st, -3rd, and -5th sublayers are guides for eyes.

Similar to the charge-ordered phase in the 25% La-doped STO, TiO_6 octahedra expand in the out-of-plane direction in the -1st, -3rd, and -5th layers (odd-numbered sublayers). Consequently, the out-of-plane t_{2g} orbitals (d_{xz}/d_{yz} orbitals) of Ti ions in the expanded octahedra become localized and energetically stable. In contrast, the -2nd and -4th layers' TiO_6 octahedra slightly contract in the out-of-plane direction, and the energy levels of the t_{2g} states are relatively unchanged. As a result, all the electrons from the La_{Sr} dopants occupy the low-lying localized d_{xz}/d_{yz} band of the odd-numbered sublayers opening the Hubbard gap, while all the t_{2g} states of the even-numbered sublayers are unoccupied. Hence, the entire 12.5% La-doped STO film becomes an insulator that consists of alternating layers of Mott insulator (odd-numbered sublayers) and band insulator (even-numbered sublayers). This charge ordering (CO) along the out-of-plane direction is consistent with that observed in the 25% La-doped STO film (Fig. 4b-d). It should be noted that, as we described in the manuscript, expanded and contracted TiO_6 octahedra in the out-of-plane direction are originated from intra-layer breathing mode (Φ_1), regardless of the doping concentration (Fig. R4a).

Fig. R4 | Schematics of the $\text{Ti}^{3+}/\text{Ti}^{4+}$ ionic ordering depending on various La-doping concentration. **a**, Charge ordering (CO) along the out-of-plane direction. Electrons are accumulated and localized due to the intra-layer breathing mode (Φ_1). **b,c**, CO along the in-plane direction. The checkerboard arrangement of $\text{Ti}^{3+}/\text{Ti}^{4+}$ ionic ordering of 25% La-doped STO (**b**). In 12.5% La-doped STO, diagonal rows of $\text{Ti}^{3+}/\text{Ti}^{4+}$ ionic ordering accompanied by La_{Sr} are formed (**c**). In **a-c**, green, purple, chartreuse, red, blue, yellow, and white spheres represent the (La, Sr), Sr, La, O atoms and Ti^{4+} , Ti^{3+} , and partially filled $\text{Ti}^{(4-\delta)+}$ cations, respectively. The yellow shades on the Ti^{3+}O_6 octahedral surface are guides for eyes.

On the other hand, we found that the CO along the in-plane direction strongly depends on La-doping concentration. In 25% La-doped STO, $\text{Ti}^{3+}/\text{Ti}^{4+}$ ionic ordering has an in-plane checkerboard arrangement (Fig. R4b), while, in 12.5% La-doped STO, $\text{Ti}^{3+}/\text{Ti}^{4+}$ ionic ordering along the in-plane direction depends on the site configuration of La_{Sr} . By evaluating the stability of the several different configurations for the $\text{Ti}^{3+}/\text{Ti}^{4+}$ ionic ordering arrangement along the in-plane direction, we found that the electrons originated from La_{Sr} tend to be localized at the nearest neighbor Ti sites of La_{Sr} (Fig. R4c). This implies that the formation of diagonal rows of $\text{Ti}^{3+}/\text{Ti}^{4+}$ ionic ordering is accompanied by La_{Sr} .

We expect that such an ordering of $\text{Ti}^{3+}/\text{Ti}^{4+}$ with La_{Sr} is a primary building block for the charge-ordered phase in the La-doped STO system. The lattice periodicity along the [110] or the $[1\bar{1}0]$ direction for the ordering of $\text{Ti}^{3+}/\text{Ti}^{4+}$ with La_{Sr} may depend on the La-doping concentration. In fact, similar types of charge-ordered stripes have been reported in commensurate [*Nature* **392**, 473–476 (1998)] and incommensurate [*Proc. Natl. Acad. Sci. U.S.A.* **115**, 1445–1450 (2018)] carrier-doped manganite. The diagonal Mn^{3+} stripes are separated by the regions of Mn^{4+} ions and create a periodic array, that is charge-ordered phase [*Nature* **392**, 473–476 (1998)]. The spacing between diagonal Mn^{3+} stripes is inversely proportional to the doping concentration. Therefore, we expect that the CO phase is not limited to a specific doping concentration such as 25% La_{Sr} , and might exist in a certain range of doping concentrations at the $\text{La}_x\text{Sr}_{1-x}\text{TiO}_3$ system.

We added discussions on this issue and additional experimental and calculation results into the revised manuscript and supporting information as below.

Main Text:

(Page 12) “In addition, we anticipate that the CO phase is not limited to a specific doping concentration such as 25% La_{Sr} , but could potentially exist in a certain range of doping concentrations in the $\text{La}_x\text{Sr}_{1-x}\text{TiO}_3$ system (see Supplementary Note 12).”

Supporting Information:

“Supplementary Note 12: Charge-ordered phase in different La-doping concentration” is added in the revised Supporting Information.

(Note 12) Fig. R2-R4 are added in the revised Supporting Information as Supplementary Fig. 22-24.

(3-1) What the activation energy from the transport data? Whether this is the same as the optical band gap?

The resistivity-temperature curve of 6 u.c. LSTO film closely follows a thermally activated adiabatic small polaron hopping (Fig. R5a) [*Sci. Rep.* **3**, 3284 (2013), *Phys. Rev. B* **92**, 035145 (2015), *Adv. Mater.* **32**, 2004490 (2020), *Isr. J. Chem.* **60**, 768-786 (2020)], and the activation energy (E_A) is estimated at 103.39 meV. In this case, the electron hops between Ti^{3+} sites and the next neighbor of Ti^{4+} sites. The temperature-dependent resistivity for the small polaron hopping (SPH) in the adiabatic regime can be expressed as:

$$\rho(T) = \rho_0 T \exp(E_A/k_B T) \quad (R1)$$

where ρ_0 is a pre-exponential factor, T is the absolute temperature, and k_B is the Boltzmann constant.

On the other hand, we obtained an optical gap of 600 meV from an onset energy (E_{Onset}) of optical conductivity spectra. In the small polaron system, the optical gap is regarded as twice the polaron binding energy (E_P) due to the two relaxation channels which can lead to the hopping transfer or on-site relaxation [*Phys. Rev. B* **92**, 035145 (2015), *Isr. J. Chem.* **60**, 768-786 (2020)]. The activation energy (E_A) is related to the polaron binding energy (E_P) and electron transfer integral (J). The transfer integral (J) is determined by tunneling process, thus depends on the wavefunction overlap or orbital overlap between adjacent sites. In addition, inter-site Coulomb repulsion (E_C) should be considered especially in highly doped systems since it disturbs electron hopping process. [*Mater. Res. Express* **1**, 046403 (2014), *Phys. Rev. B* **92**, 035145 (2015)]. Overall, the relationship of E_A , E_P , J , and E_C in the SPH model can be expressed as [*Phys. Rev. B* **92**, 035145 (2015)]:

$$E_A = 1/2 E_P - J + E_C. \quad (R2)$$

As can be shown from equation (R2), the activation energy obtained from transport properties typically tends to be less than a quarter of the optical gap measured by optical spectroscopy. This is also consistent with our case.

Fig. R5. | **a**, Temperature versus resistivity curve of 6 u.c. thick of LSTO film plotted as the adiabatic small polaron hopping model. The blue line is a linear fitting curve to estimate activation energy (E_A). **b**, Optical conductivity spectra of the 6 u.c. thick LSTO films. The magenta line is a linear extrapolation of the low energy shoulder of a fit curve to determine an onset energy (E_{Onset}).

We added the above discussion and additional data to the revised manuscript and supporting information as below.

Main Text:

(Page 6) “A detailed analysis of transport properties is described in Supplementary Note 3.”

(Page 7) “It should be noted that the incoherent mode near 1.3 eV in our LSTO film cannot be explained by quasi-classical polarons. The quasi-classical polaron state in doped SrTiO₃ is reported to form below 0.5 eV^{27,28}, which is significantly lower than the energy level observed in our case. However, in ordered phases such as the CO and orbital ordering (OO), the inter-site hopping of electrons is often disrupted by Coulomb repulsion, leading to an increase in the optical gap²⁹. Previous reports have indicated that transition metal oxide ultrathin film undergoing a correlated induced MIT exhibit an absorption near 1.5 eV^{30,31}. Based on the findings, we speculate that the adsorption near 1.3 eV in ultrathin LSTO film is likely attributed to the correlated induced MIT and is reminiscent of the CO phase.”

Caption of Figure 2d:

“The black dotted line is an extension of temperature-dependent resistivity curve, following an adiabatic small polaron hopping (See Supplementary Note 3).”

Reference #29 is added.

Supporting Information:

“Supplementary Note 3: Transport properties of LSTO films” is added in the revised Supporting Information.

(Note 3) Fig. R5a is added in the revised Supporting Information as Supplementary Fig. 5.

(Note 5) “The incoherent mode near 1.3 eV, observed in our LSTO films (Fig. 2e), was not present at the SrTiO₃ substrate (Supplementary Fig. 7). This implies that the incoherent spectral peaks are the intrinsic behavior of the LSTO films.”

(Note 5) “To determine an optical gap, we extracted an onset energy (E_{Onset}) of optical conductivity spectra. Extrapolating the low energy shoulder of the incoherent mode peak near 1.3 eV of semiconducting 6 u.c. thick LSTO film (Supplementary Fig. 8), we obtained the onset energy (E_{Onset}) of 600 meV. In the small polaron system, the optical gap is regarded as twice the polaron binding energy (E_p) due to the two relaxation channels which can lead to the hopping transfer or on-site relaxation^{5,7}. The activation energy (E_A) is related to the polaron binding energy (E_p) and electron transfer integral (J). The transfer integral (J) is determined by tunneling process, thus depends on the wavefunction overlap or orbital overlap between adjacent sites. In addition, intersite Coulomb repulsion (E_C) should be considered especially in highly doped systems since it disturbs electron hopping^{5,9}. The relationship of E_A , E_p , J , and E_C in the SPH model can be expressed as⁵:

$$E_A = 1/2E_p - J + E_C. \quad (2)”$$

(Note 5) Fig. R5b is added into the revised Supporting Information as Supplementary Fig. 8.

(3-2) Also, the 8 uc sample is also very exotic with extremely large RRR. Can authors comment on this?

As reviewer pointed out, 8 u.c. thick LSTO film shows a very large residual resistivity ratio (RRR). We attribute that this is originated from substrate conduction. The carrier spreading can evoke a metallic response from the substrate even when the substrate is originally insulating. The layer-resolved density of state (LDOS) derived from DFT calculation for the 8 u.c. thick LSTO film shows the metallic state with the occupied conduction bands on the -8th and -9th TiO₂ sublayers, which is in the STO substrate of the LSTO/STO slab structure. This indicates that the electrons originating from La_{Sr} dopants, which are located on the -6.5th and -7.5th (La, Sr)O sublayer, were transferred to the STO substrate due to the formation of LSTO/STO junctions [*Phys. Rev. X* **7**, 011023 (2017)]. Since ionized impurity scattering center such as La_{Sr} dopant does not exist in STO substrate, these electrons can exhibit high mobility at low temperatures, resulting in large RRR. It has been reported that low temperature electron mobility of LSTO thin films increased with the decrease in La-doping concentration, leading to a large RRR [*Nat. Mater.* **9**, 482-484 (2010)].

Fig. R6. | **a**, Schematic representation of the atomic structure of the 8 u.c. thick LSTO film on STO (001) surface. A copy of Supplementary Fig. 18a. **b**, The layer-resolved DOS projected onto the Ti 3d t_{2g} orbitals of the 8 u.c. thick LSTO film. A copy of Supplementary Fig. 18c. **c**, Zoom-in of layer-resolved DOS from -6th to -9th TiO₂ sublayer.

We added the above discussion to the revised supporting information as below.

Supporting Information:

(Note 8) “When $t = 8$ u.c., the layer-resolved density of state (LDOS) shows the metallic state with the occupied conduction bands on the -8th and -9th layers, which is in the STO substrate (Supplementary Fig. 18). This indicates that the electrons originating from La_{Sr} dopants, which is located on the -6.5th and -7.5th sublayer, were transferred to the STO substrate due to the

formation of LSTO/STO junction¹³. The carrier spreading can evoke a metallic response from a substrate even when the substrate was insulating. Since ionized impurity scattering center such as La_{Sr} dopant does not exist in STO substrate, these electrons can exhibit high mobility at low temperatures, resulting in an extremely large residual resistivity ratio (RRR) (Fig. 2d). It has been reported that, low temperature electron mobility of LSTO thin films increased with the decrease in La-doping concentration, leading to a large RRR¹⁴. Therefore, we attribute that the large RRR of the 8 u.c. thick LSTO thin film originates from substrate conduction.”

(4) It is not clear how the authors modeled the surface state during their calculations.

We understand the reviewer's concern and thanks for the constructive comments.

The prevailing technique for modeling surface structures in DFT is a repeated slab approach. In this method, by the introduction of a vacuum region within the supercell, the translational symmetry along the direction perpendicular to the surface is broken [*Npj Comput. Mater.* **7**, 58 (2021)]. Based on the experimental observation of the LSTO films' $a^0a^0c^{\cdot}$ Glazer's notation pattern as well as the STO substrate [*Chem. Mater.* **4**, 346–353 (1992), *Appl. Phys. Lett.* **115**, 161601 (2019)], we construct the candidate structures of the LSTO/STO heterogeneous slab by La atom substituting a Sr atom site of the STO slabs.

To investigate LSTO film on the STO substrate with varying film thickness, we constructed the STO slabs that have mirror symmetry TiO_2 -terminated surfaces with various thicknesses (including 8.5, 10.5, 12.5, and 14.5 unit cell layers). La atoms were substituted in one of four Sr atom sites on the SrO sublayers to construct LSTO film which corresponds to 25% La-doping concentrations. While 6 bottom SrO sublayers were not substituted to simulate 6.5 u.c. of the undoped TiO_2 -terminated STO substrates, the LSTO films with various thickness, t ($t = 2, 4, 6, \text{ and } 8$ u.c.) were placed on the top of the STO substrate (Supplementary Fig. 9). A vacuum, length of $4c$ ($4c = 1.59464$ nm), was included in the slab structures, and the in-plane directions, $2a \times 2a$ ($2a = 0.79174$ nm) cell was used. Where a and c were those of the optimized values of the STO bulk.

The atomic structures of the constructed LSTO/STO slab were optimized by relaxing the atomic positions with the atoms on the 3 bottom layers of the substrate fixed. When we performed the calculation using slab structure, the dipole correction was used in the z -direction to alleviate the unintended external electric field resulting from the image supercell.

Note that the remaining terms (Φ'_{rem}) of distortion mainly come from the atomic displacements on the surface layer (Supplementary Fig. 15) when the lattice modulation of LSTO/STO slab is decomposed into the 4 representative octahedral distortion modes: the inter-layer breathing mode (Φ'_1), intra-layer breathing mode (Φ'_2), Jahn-Teller type distortion mode (Φ'_3), and the antiferrodistortive rotation (Φ'_z). These surface atomic displacements result in a downward atomic rumpling on the 0th TiO_2 sublayer (top surface) and an upward atomic rumpling on the -0.5th (La, Sr)O sublayer (Fig. R7). This DFT calculated atomic rumpling behavior of the LSTO film surface is also reported in undoped TiO_2 -terminated STO (001) surface [*J. Phys. Chem. C* **123**, 8086–8091 (2019)].

As the reviewer suggested, we added a brief explanation into the revised supporting information as below.

Supporting Information:

(Note 8) “Based on the experimental observation of the LSTO films' $a^0a^0c^{\cdot}$ Glazer's notation

pattern as well as the STO substrate^{1,2}, we constructed the candidate structures of the LSTO/STO heterogeneous slab by La atom substituting a Sr atom site of the STO slabs.”

(Note 8) “Using the obtained atomic structure of STO bulk, we constructed a TiO₂-terminated STO mirror symmetric slabs with various thicknesses (including 8.5, 10.5, 12.5, and 14.5 unit cell layer) to investigate LSTO film on the STO substrate with varying film thickness. La atoms were substituted in one of four Sr atom sites on the SrO layers to construct LSTO film which corresponds to 25% La-doping concentrations. While 6 bottom SrO sublayers were not substituted to simulate 6.5 u.c. of the undoped TiO₂-terminated STO for the substrates, the LSTO films with various thickness, t ($t = 2, 4, 6, \text{ and } 8$ u.c.), were placed on the top of the STO substrate (Supplementary Fig. 12).”

(Note 8) “When we performed the calculation using slab structure, the dipole correction was used in the z -direction to alleviate the unintended external electric field resulting from the image supercell.”

(Note 8) “These surface atomic displacements result in a downward atomic rumpling on the 0th TiO₂ sublayer (top surface) and an upward atomic rumpling on the -0.5th (La, Sr)O layer. This DFT calculated rumpling behavior of the LSTO film surface is also reported in undoped TiO₂-terminated STO (001) surface (see Supplementary Note 9).”

Reviewer #2 (Remarks to the Author)

- Comments:

The paper reports the dead layer formation in the La-doped SrTiO₃ (LSTO) thin film deposited on a SrTiO₃ substrate. The high electrical resistance observed in the film below the critical thickness is attributed to charge ordering. From the COBRA and STEM-EELS measurements, the authors argue on the structures of the heterostructures.

I do not recommend publishing in Nature Communications because the study seems incomplete. If the surface triggers the metal-insulator transition, it is essential to determine the surface structure experimentally. In particular, it is critical to clarify whether the surface is reconstructed. Investigating the in-plane structures at the atomic level by RHEED, STM, or other methods is recommended. In addition, the discussion is weak because the electronic states are discussed only in terms of electron transport properties. Probing the electronic states near the Fermi level directly using photoelectron spectroscopy is desired to connect with calculation results. minor issue: Some of the paragraphs are too long to read. Finally, information should be added on whether the transport property measurements were conducted in non-exposed atmospheres. It is known that oxygen adsorbs on the film surface in ambient air and induces metal-to-insulator transition. Such an effect is also possible.

We greatly thank the reviewer for constructive and valuable comments, and suggestion. Following the comments and suggestion, we have made extensive experiments and further analysis along with theoretical calculations including synchrotron-based X-ray spectroscopy measurements, RHEED, further analysis of COBRA, and DFT calculations. We have been able to improve our manuscript significantly in the revision from the further experiments and analysis. In the pages that follow, we provide our response to each of the questions, in order.

(1) If the surface triggers the metal-insulator transition, it is essential to determine the surface structure experimentally. In particular, it is critical to clarify whether the surface is reconstructed. Investigating the in-plane structures at the atomic level by RHEED, STM, or other methods is recommended.

We agree with the reviewer that experimentally determining surface structure will strongly support our argument in this manuscript. Therefore, we performed additional analysis on the topmost surfaces of LSTO films obtained by COBRA and DFT calculation, and RHEED experiment to investigate the surface structure.

We first examined the detailed atomic structure and surface atomic distortion of ultrathin La-doped STO thin films by complementary Coherent Bragg rods analysis (COBRA) studies and density functional theory (DFT) calculations (Fig. R7). The surface distortion is usually initiated by the termination of a crystalline material, i.e., dangling bonds at the topmost surface. According to Pauling's rule [*J. Am. Chem. Soc.* **51**, 1010–1026 (1929), *The nature of the chemical bond and the structure of molecules and crystals; an introduction to modern structural chemistry* (3rd ed.) 543–562 (Cornell University Press, 1960)], an equilibrium cation-anion bond length strongly depends on the coordination number. Due to the decreased

coordination number of the Ti and O atoms (TiO_6 octahedra to TiO_5 pyramid and $\text{Sr}_4\text{Ti}_2\text{O}$ octahedra to $\text{Sr}_2\text{Ti}_2\text{O}$ polyhedra) on the TiO_2 terminated SrTiO_3 surface, shortened Ti-O and Sr-O equilibrium bond lengths result in downward and upward rumpling on the 0th TiO_2 sublayer (top surface) and -0.5th SrO sublayer, respectively [*J. Phys. Chem. C* **123**, 8086–8091 (2019)]. Note that these surface atomic rumpling in the TiO_2 terminated La-doped STO film is well described in our DFT calculations.

To confirm this atomic rumpling behavior experimentally, we carried out synchrotron surface X-ray diffraction measurements and performed COBRA analysis of an 8 u.c. thick LSTO thin film. The complete atomic structures of each unit cell of the LSTO thin film including the surface unit cell in focus can be obtained from COBRA analysis. Figure R7b shows the COBRA derived full electron density map with 2D vertical slices on the (110) atomic plane and (200) atomic plane, respectively. The slice along the (200) atomic plane cuts through the Ti, apical oxygen, and equatorial oxygen sites. Therefore, we can quantitatively determine the Ti-O atomic rumpling ($\delta_{\text{Ti-O}}$) magnitude. Likewise, we identify (La, Sr)-O atomic rumpling ($\delta_{(\text{La, Sr})-\text{O}}$) magnitude using the slice along the (110) atomic plane cutting through the (La, Sr), and apical oxygen sites. The slices along the (200) and (110) atomic planes reveal a downward rumpling on the 0th TiO_2 sublayer (top surface) and upward rumpling on the -0.5th (La, Sr)O sublayer. Therefore, we conclude that experimentally determined surface structures such as atomic rumpling behavior at the surface layer are in good agreement with those obtained from the DFT calculations.

Fig. R7 | **a**, Calculated atomic structure near the surface of the 8 u.c. thick LSTO film on STO substrate. **b**, A 2D vertical slice through the 3D electron density map along the (200) and (110) atomic plane for an 8 u.c. thick LSTO film. (La, Sr) and equatorial oxygen is not visible along the (200) and (110) atomic plane, respectively. **c-d**, Atomic rumpling of the relaxed (100) surfaces, as a determination method: DFT and COBRA. Ti-O atomic rumpling ($\delta_{\text{Ti-O}}$) on the 0th sublayer (c) and (La, Sr)-O atomic rumpling ($\delta_{(\text{La, Sr})\text{-O}}$) on the -0.5th sublayer (d).

Next, it is important to determine whether surface reconstruction contributes to the emergence of the charge-ordered phase in La-doped STO thin film. Depending on the thermodynamic condition including temperature and pressure, several phases of SrTiO₃ (001) surface have been reported such as (1×1), (2×1), (2×2), *c*(4×2), (4×4), *c*(4×4), *c*(6×2), (6×2), ($\sqrt{5}\times\sqrt{5}$)-R26.6° and ($\sqrt{13}\times\sqrt{13}$)-R33.7° structure, respectively [*Springer Handbook of Surface Science* 169–185 (Springer Press, 2020)]. Typically, a reconstructed surface on STO is obtained through thermal annealing under ultra-high vacuum at relatively higher temperatures (Table R1). It should be noted that La-doped STO film was grown at a temperature of 650 °C (923.15 K) and 10⁻⁵ Torr of oxygen/ozone mixture, which is a relatively lower temperature and higher oxidation condition than that required for surface reconstruction (Table R1). In addition, before

the film growth, we performed short re-etching procedure using buffered-HF to maintain a TiO₂ termination up to growth temperature (i.e., 650 °C) [*Appl. Phys. Lett.* **85**, 272–274 (2004)].

Nonetheless, to examine the potential presence of an in-plane reconstruction phase on the surface, we investigated the surface structure using reflection high-energy electron diffraction (RHEED) before and after LSTO film growth. We monitored the RHEED pattern of the pre-growth STO (001) surface and the post-growth LSTO thin film surface with [100] beam incidences. No additional superstructure spot was observed before or after LSTO film growth. Thus, we conclude that the surface of our LSTO thin film remains in an unreconstructed state, which is also reported in the following references [*Physica C* **229**, 1-11 (1994), *Appl. Phys. Lett.* **105**, 191901 (2014)].

Struct.	Sample	Temperature (K)										Time (min/h)	Method		
		800	900	1000	1100	1200	1300	1400	1500	1600	1700				
(1×1)	Fractured				1200 K UHV										LEED
	Ar ⁺		1100 K UHV											60	LEED
	Ar ⁺		1400 K O ₂ H ₂ + 1300 K UHV												LEED
	Polished				870–1070 K UHV									2–120	LEED, STM
	Polished					1020–1170 K 10 ⁻⁵ /0.15 mbar O ₂								10	LEED, STM
	Ar ⁺ Bt				900 K (UHV + 1×10 ⁻⁶ O ₂)										LEED, SXRD
	BHF					870–1270 K 5×10 ⁻⁶ /1×10 ⁻⁵ mbar O ₂								30–60	LEED, RHEED, MEIS
	BHF				870 K UHV									30	LEED
	BHF					1220 K UHV (2×1+2×2+1×1)								60	XRD
(2×1)	Ar ⁺				1100 K 10 ⁻⁵ mbar O ₂ and UHV (2×1+2×2+1×1)									60	LEED
	Polished					1020–1170 K UHV								20–120	LEED, RHEED, STM
	(1×1)					1220 K 5×10 ⁻⁷ mbar H ₂								120	LEED, STM
	BHF					870–1070 K UHV								30	LEED, STM
	BHF					1220 K UHV (2×1+2×2+1×1)								60	XRD
	BHF					1170 K 1×10 ⁻² mbar O ₂								30	RHEED
	Ion milling					1220–1270 K O ₂ flow								0.5–5 h	HRTEM
(2×2)	Ar ⁺				1100 K 1×10 ⁻⁵ mbar O ₂ and UHV (2×1+2×2+1×1)									60	LEED
	Polished					1270–1470 K UHV								2–20	LEED, STM
	Polished		970 K			1×10 ⁻⁷ mbar O ₂								Hours	LEED
	BHF					1270 K UHV [2×2 + c(4×4)]								20	STM
	BHF					1220 K UHV (2×1+2×2+1×1)								60	XRD
	Ar ⁺ Bt					1070 K UHV								5 h	STM
	BHF					870–1270 K UHV								20–120	STM, LEED
c(4×2)	Polished					1220 K UHV + H ₂ 5×10 ⁻⁷ and 10 ⁻⁵ mbar								120	LEED, STM
	BHF + Ar ⁺					1470 K UHV								15	LEED, STM
	BHF + Ar ⁺					1130 K UHV								5	LEED, STM
	Ion milling					1100–1200 K flowing O ₂								0.5–5 h	TEM
	BHF + Ar ⁺					1470 K UHV								15	STM
	Ar ⁺					1120 K 10 ⁻⁶ O ₂								Cycle	LEED, STM
	BHF					1270 K flowing O ₂								30	LEED
(4×4)	BHF					c(4×4) to 1450 K = 4×4 + √5×√5								Flash	STM
	BHF		1170			1670 K [(2×1)]								> 30	LEED, STM
c(4×4) UHV dots	BHF					1370 K								20	STM
	BHF					2×2 to 1300 K = c(4×4)								120	STM
	Polished					1270 K								60	STM
	Polished					1070–1370 K flow O ₂								15 h	
c(6×2) O ₂ flow	Polished					Traces c(6×2) in √13×√13								3–5 h	LEED, STM
	Polished					220–1370K, 3 – 5 h, Fl. O ₂ 1220 K 5×10 ⁻⁷ H ₂ or UHV								3–5 h	LEED, STM
	Ion milled					1320–1370 K O ₂ flow								2–5 h	TEM, XRD, STM
	Ion milled					1320 K or 1320–1470 K O ₂ flow								10 h	TEM
√5 ×√5 UHV	Polished					1470 K								2	RHEED, STM
	Polished,					15 h 1170 K 2 min 1470 K								2	RHEED, STM
						1455 K								2	STM
						1100 K								120	LEED
	BHF					1470 K								2	STM, NC-AFM
	BHF					1470 K								Flash	STM
	Polished					1670 K								80	LEED, STM
√13 ×√13	Polished					1070–1370 K flow O ₂ (traces c(6×2))								15 h	RHEED
	BHF					1320 K O ₂ flow – air stable								5 h	TEM
	BHF					1270 K flowing O ₂								10 h	LEED
	BHF					1520 K (after √5 × √5)								Flash	STM
	BHF					1120 K 10 ⁻⁵ mbar O ₂								40	RHEED, STM

Growth temp. of LSTO thin film, 650 °C

Table R1 | Reconstructions of the SrTiO₃ (001) surface by thermal annealing with different temperatures and times. Surface treatment methodology, annealing temperature, annealing pressure, and annealing times (numbers without indication are minutes) are given. *Horizontal red bars* show annealing temperature ranges. Adapted from Fig. 6.16 in *Springer Handbook of Surface Science* (Springer Press, 2020).

Fig. R8 | RHEED patterns for LSTO thin film growth obtained at room temperature along the [100] azimuths. a, RHEED pattern of the STO (001) surface after the annealing process. The STO substrate was etched 45 sec using buffered-HF and annealed at 1100 °C for 6 hours. Before the film growth, the second chemical etching procedure using buffered-HF for 10 sec was performed to stabilize the TiO₂-terminated surface of the STO substrates [*Appl. Phys. Lett.* **85**, 272–274 (2004)]. **b**, RHEED pattern of the LSTO thin film surface after growth. The LSTO thin film is grown under 10⁻⁵ Torr of oxygen/ozone mixture and at 650 °C.

To clarify this issue further, we added brief explanations in the main text, methods section of the revised manuscript and supporting information as follows.

Main Text:

(Page 11) “It should be noted that we could not find any evidence of the reconstructed structure on the surface of the LSTO thin film (see Supplementary Note 9).”

(Page 12) “We first confirm that surface distortion, which originates from the dangling bond in the perovskite oxide thin film⁴², through surface X-ray diffraction measurements and performed COBRA analysis (see Supplementary Note 9). The slices along the (200) and (110) atomic planes reveal a downward rumpling on the 0th TiO₂ sublayer (top surface) and upward rumpling on the -0.5th (La, Sr)O sublayer (Supplementary Fig. 20). Note that experimentally determined surface structures such as atomic rumpling behavior at the surface layer is in a good agreement with those obtained from the DFT calculations.”

Reference #42 is added.

Methods Section:

(Page 17) “Before the film growth, we performed the second chemical etching procedure using buffered-HF for 10 sec, to stabilize TiO₂-terminated surface of the STO substrate⁴⁸.”

Reference #48 is added.

Supporting Information:

“Supplementary Note 9: Verification of surface structure” is added in the revised Supporting Information.

(Note 9) Fig. R7 is added in the revised Supporting Information as Supplementary Fig. 20.

(Note 9) Fig. R8 is added in the revised Supporting Information as Supplementary Fig. 19.

(2) In addition, the discussion is weak because the electronic states are discussed only in terms of electron transport properties. Probing the electronic states near the Fermi level directly using photoelectron spectroscopy is desired to connect with calculation results.

We appreciate the reviewer's concern and the constructive comments. As pointed out by the reviewer, additional experiments were performed to obtain further information on the electronic structure of the charge-ordered phase in ultrathin LSTO films. Considering the availability of synchrotron-based X-ray spectroscopy measurement in our situation, we decided to perform synchrotron-based soft X-ray absorption spectroscopy (XAS) to probe the electronic states near the Fermi level, that is complementary to photoelectron spectroscopy, and investigate the electronic structure of the conduction band for the LSTO films since soft X-ray absorption spectroscopic can directly probe the unoccupied electronic state with chemical and orbital sensitivity [*Nat. Commun.* **5**, 4291 (2014), *Nano Lett.* **17**, 5458–5463 (2017), *ACS Nano* **17**, 14814–14821 (2023)]. Notably, XAS results show that the lowest unoccupied state at conduction band of the ultrathin La-doped STO film consists of the d_{xy} orbital. This is in good agreement with the electronic states predicted by DFT calculations, as shown in Figure 4d and Figure R9a.

Fig. R9 | Orbital structure of LSTO thin films. a, The calculated density of states of 6 u.c. thick LSTO film projected onto the Ti $3d t_{2g}$ orbitals. Purple and green lines represent the DOS

projected onto the Ti d_{xy} and $d_{xz}+d_{yz}$, respectively. The midpoint of the energies of the lowest unoccupied state and the highest occupied state is set to be zero and represented by the vertical line. The lowest unoccupied state consists of d_{xy} orbital originated from even-numbered sublayers (Fig. 4d) is at $E \sim 0.5$ eV. **b-d**, Characteristic X-ray absorption spectroscopy (XAS) at Ti- $L_{2,3}$ edge. **b**, X-ray linear dichroism (XLD) spectra ($I \parallel c - I \parallel ab$) of the LSTO film for $t = 30$ u.c. (blue line) and 6 u.c. (red line). **c,d**, Normalized Ti- $L_{2,3}$ edge XAS spectra of LSTO films for $t = 30$ u.c. (c), and for $t = 6$ u.c. (d), respectively.

The XAS measurement has been performed at the 2A beamline of the Pohang Accelerator Laboratory. XAS spectra were collected at the Ti- $L_{2,3}$ absorption edge with photon polarization $E \parallel ab$ (parallel to the sample plane) and $E \parallel c$ (almost perpendicular to it) in total electron yield mode (surface sensitive probing). In the grazing incidence experimental geometry, linearly polarized X-rays with the electric field orientation parallel to the c -axis ($E \parallel c$) preferentially probe the unoccupied (d_{xz} , d_{yz}) states. X-rays with their electric-field orientation perpendicular to the c -axis ($E \parallel ab$) are more sensitive in probing the state with d_{xy} orbitals. The difference between the spectra of the two polarization configurations (dichroism) is pronounced for 6 u.c. film, while the difference is smaller for thicker films, 30 u.c.. The X-ray linear dichroism (XLD) signal is derived from the intensity difference ($I \parallel c - I \parallel ab$) of normalized XAS spectra, as shown in Figure R9b. The XLD intensity decreases with increasing the thickness, implying that the orbital degeneracy is lifted by reduction of film thickness. In the case of thinner films (i.e., an insulating film), in-plane orbital (d_{xy}) is found to be lowest unoccupied state of conduction band since the energy of $L_{3:t_{2g}}$ peak obtained by $E \parallel c$ is much higher than that obtained by $E \parallel ab$ (Fig. R9b-d).

It should be noted that such an electronic structure near the conduction band consisting of d_{xy} state is consistent with the DFT calculations. As shown in Figure R9a, the orbital projected density of states of the charge-ordered 6 u.c. thick LSTO film shows predominant Ti d_{xy} orbital polarization in the conduction band edge. This is originated from inter-layer breathing mode (Φ_1) that a contracted TiO_6 oxygen octahedra in the $[001]$ direction within the -2nd and -4th layers, leading to stabilize d_{xy} states as shown in the layer projected DOS (Fig. 4d). On the other hand, in the -1st, -3rd, and -5th layers (odd-numbered sublayers), where the TiO_6 octahedron expands in the out-of-plane direction, d_{xz} and d_{yz} orbitals become localized and energetically favorable. Therefore, the electrons occupied at the low-lying localized d_{xz}/d_{yz} band open the Hubbard gap while other t_{2g} states at the even-numbered sublayers remain unoccupied states. Therefore, we conclude that electronic structures explored by the X-ray linear dichroism experiment are in good agreement with those obtained from the DFT calculations.

We added the above discussion and additional experimental results to the revised manuscript as follows. Fig. R9a,b are also added in the revised manuscript as Fig. 4f,g.

Main Text:

(Page 10) “To further support validity of DFT predicted electronic structures for the charge-

ordered phase in LSTO film, we performed synchrotron-based X-ray absorption spectroscopic (XAS) since XAS can directly probe the unoccupied states (i.e., conduction band) near the Fermi energy with chemical and orbital sensitivity³⁴⁻³⁶. As shown in Figure 4f, DOS of the charge-ordered LSTO film ($t = 6$ u.c.) obtained from DFT calculation shows predominant Ti d_{xy} orbital polarization in the lowest unoccupied state (i.e., conduction band edge). This originates from the contracted TiO_6 oxygen octahedra in the [001] direction within the -2nd and -4th layers (Fig. 4b-d) in the inter-layer breathing mode (Φ_1). We collected XAS spectra at the Ti- $L_{2,3}$ absorption edge with photon polarization $E \parallel ab$ (parallel to the sample plane) and $E \parallel c$ (almost perpendicular to it) in total electron yield mode. The difference between the spectra of the two polarization configurations is pronounced when $t = 6$ u.c., while the difference is smaller for thicker films ($t = 30$ u.c.) (Fig. 4g). In the case of thinner films (i.e., charge-ordered insulating film), in-plane orbital (d_{xy}) is found to be lowest unoccupied state of conduction band since the energy of $L_3:t_{2g}$ peak obtained by $E \parallel c$ is much higher than that obtained by $E \parallel ab$ (Fig. 4g). These results reveal that the lowest unoccupied state of the ultrathin LSTO film consists of d_{xy} orbital, which is consistent with those obtained from the DFT calculations of the charge-ordered phase in LSTO film (Fig. 4f).”

Reference #34, #36 are added.

Methods Section:

(Page 18) “The XAS and XLD measurement has been performed at the 2A beamline of the Pohang Accelerator Laboratory at 300 K, in total electron yield mode. The spectra were measured at the Ti- $L_{2,3}$ edge for the two polarizations (in-plane and out-of-plane). The direction of the soft X-ray polarization vector E was changed by 90° to obtain the in-plane ($E \parallel ab$) and the out-of-plane ($E \parallel c$) orbital responses, while the angle of incidence was 70° with respect to the surface normal direction. Soft X-rays with their electric-field orientation perpendicular to the c -axis ($E \parallel ab$) are more sensitive in probing the state with d_{xy} orbitals. The X-ray linear dichroism (XLD) signals were derived from the intensity difference ($I \parallel c - I \parallel ab$) of normalized XAS spectra.”

(3) minor issue: Some of the paragraphs are too long to read.

We agree with the reviewer. We reorganized some of the paragraphs to increase the readability as follows.

Main Text:

(Page 5) “The contrast inverse annular bright-field (ABF) STEM image shows that LSTO/STO interfaces are atomically sharp and do not have any misfit dislocations (Fig. 2c).

¶To explore the evidence of phase transition induced by surface distortion, resistivity is measured as a function of temperature for LSTO films with various thicknesses.”

(Page 8) “Note that the two-sublattice modulation of TiO_6 octahedron height (Fig. 3c) corresponds to inter-layer breathing mode (Φ_1) (Fig. 1c).

¶The electron density is also examined by STEM-EELS analysis.”

(Page 9) “The calculated layer-resolved density of states (DOS) shows the periodic modulation of the electron density (Fig. 4d).

¶We further analyze the obtained atomic and electronic structure in detail, and the lattice modulation by decomposing it into the 4 representative octahedral distortion modes:”

(Page 10) “The 2 and 4 u.c. thick LSTO film ($t = 2, 4$ u.c.) showed similar results to that of 6 u.c. thick film (Supplementary Fig. 17).

¶For LSTO films thicker than the characteristic length ($t > 6$ u.c.), the top 6 layers with the CO state are sitting on top of the metallic layers which do not have such a distortion.”

(4) Finally, information should be added on whether the transport property measurements were conducted in non-exposed atmospheres. It is known that oxygen adsorbs on the film surface in ambient air and induces metal-to-insulator transition. Such an effect is also possible.

This is another interesting and insightful comment. We first would like to mention that our transport properties are measured by a physical-property measurement system (PPMS) under a rough vacuum of a 10^{-3} Torr. Information regarding the transport property measurement has been included in the revised manuscript as shown below.

We also agree with the reviewer's comment that oxygen adsorption on the film surface can influence the electronic structure and consequent transport properties of the transition metal oxide heterostructures. When the surface of metallic film is decorated with oxygen adsorbates, the occupation of d orbital can be decreased. This implies that the oxygen adsorbates drain electrons from the film surface, leading to the formation of a space charge region near the top surface region. However, it has been known that this effect is only confined to the surface regime, typically within 1 or 2 u.c. from the surface [*Adv. Electron. Mater.* **8**, 2101006 (2022)]. Considering relatively thicker critical thickness for MIT in ultrathin LSTO film than the thickness of an electrical dead layer originated from oxygen adsorbates [*Adv. Electron. Mater.* **8**, 2101006 (2022)], we believe that oxygen adsorption only plays a minor role in the electrical properties of the La-doped STO system.

To clarify the issue raised by the reviewer, we added a brief explanation into the revised manuscript and supporting information (Supplementary Note 10) as follows.

Methods Section:

(Page 17) "The electrical transport was measured using a physical-property measurement system (PPMS) under a rough vacuum of a 10^{-3} Torr. Au/Pt were used as contact metals for van der Pauw geometry."

Main Text:

(Page 11) "Another possible mechanism to explain surface-driven MIT is the electron draining originated from oxygen adsorbates on the film surface⁴¹. However, it has been reported that such an effect is limited to 1 or 2 u.c. of the topmost surface⁴¹. Considering the relatively thicker critical thickness for the MIT in LSTO thin film than the thickness of an electrical dead layer originated from oxygen adsorbates⁴¹, we can exclude the effect of oxygen adsorbates as the origin of the thickness-driven MIT in the LSTO film (Supplementary Note 10)."

Reference #41 is added.

Supporting Information:

"Supplementary Note 10: Effect of oxygen adsorption on the film surface for electrical properties" is added in the revised Supporting Information.

REVIEWERS' COMMENTS

Reviewer #1 (Remarks to the Author):

The authors have addressed nicely the comments from the referee, and now the paper is ready to be published.